# LAPLACIAN FLOWS FOR POLICY LEARNING FROM EXPERIENCE

**Xingrui Gu**
Electrical Engineering and Computer Sciences
University of California, Berkeley
xingrui_gu@berkeley.edu

**Chuyi Jiang**
Department of Electrical Engineering
Columbia University
cj2792@columbia.edu

## ABSTRACT

Many learning and decision-making systems output conditional distributions rather than point predictions, yet are trained via locally "reasonable" myopic gradient updates that implicitly assume their composition remains globally stable and feasible. In RL (policy gradient/actor–critic) and LLMs (cross-entropy), drift is typically controlled by KL/Fisher trust regions, which need not reflect the true behavioral scale of policy change, so small per-step moves can accumulate into large transport-scale shifts that break stability, long-horizon evidence integration, and robustness (like a millimeter map error causing a catastrophic fall in physical space). We propose the *Policy Laplacian Trace* (PLT): retrieved historical policies define an OT-induced local graph, and each update solves a variational OT+KL proximal step coupling a Wasserstein barycenter term with KL regularization, yielding experience-induced Laplacian smoothing of task-gradient drift. Geometrically, PLT connects to Laplace learning in Wasserstein space: its discrete graph energy approximates a $p$-Dirichlet/Laplace–Beltrami energy on the realizable policy subset. Empirically, PLT is plug-and-play and improves PPO/MAPPO stability, sample efficiency, and robustness under controlled shifts, and strengthens LLM-as-policy performance on counterfactual trust, long-range factual recall, and few-shot novel-category learning across GPT-family models, while maintaining or improving base performance and calibration.

## 1 INTRODUCTION

Many modern learning and decision-making systems are best viewed as conditional distribution learners (Zhao et al., 2023; Wang et al., 2020): given an input $x$ (state, context, or prompt), a model outputs a conditional measure on the action space $\mathcal{A}$, $\pi_\theta(\cdot \mid x) \in \mathcal{P}(\mathcal{A})$ (Doob, 1941; Ash & Doléans-Dade, 2000).A natural way to quantify policy change is by tracking the movement of probability mass in action space, which can be formalized by endowing $\mathcal{A}$ with a ground metric and inducing a 2-Wasserstein geometry $W_2$ on $\mathcal{P}_2(\mathcal{A})$. (Villani et al., 2008).

A key mismatch is that updates are controlled in parameter/KL space and behavior is governed by action-space transport. Assuming extra regularity, small KL can not control $W_2$, so KL trust regions can still allow large transport shifts which degrades stability and generalization (Schulman et al., 2017; Achiam et al., 2017; Peyré et al., 2019; Shah et al., 2025). The action space $a \in \mathcal{A}$ carries task semantics through a ground cost $c(a, a')$; in continuous control, small and large action displacements (e.g. $a_t \to a_{t+1}$ and $a_t \to a_{t+100}$) incur fundamentally different costs, KL-based divergences are geometry-mismatched and cannot distinguish local from long-range probability shifts. In contrast, Wasserstein distances measure transport under $c(a, a')$. When behavioral drift is governed by action-space displacement, constraining $W_2$ better matches the operational scale of policy change than constraining KL alone (See counterexample in Appendix F.2). This mismatch manifests as abrupt behavioral changes, instability, brittle generalization, and pronounced robustness failures under distribution shift, suggesting that much of policy learning is effectively geometry-mismatched to the global transport structure between policies as probability measures (Villani et al., 2008; Fujimoto et al., 2019; Kumar et al., 2019; 2020).

Under this lens, a finite-dimensional realizable family $\{\mu_\theta\}_{\theta \in \Theta} \subset \mathcal{P}_2(\mathcal{A})$ (with $\Theta \subset \mathbb{R}^d$) can be viewed as a low-dimensional subset, often idealized as a submanifold, embedded in Wasserstein space, so optimization implicitly traces a trajectory on a Wasserstein policy manifold. Importantly, unlike the data manifold hypothesis (Bengio et al., 2013; Fefferman et al., 2016; Narayanan & Niyogi, 2009), the relevant geometry here concerns policies rather than observations, and it is not a priori but experience-induced: the training trajectory $\{\pi_{\theta_t}(\cdot \mid x)\}_t$ provides samples of the realizable policy set. Recent progress in Laplacian-based learning on Wasserstein spaces offers a concrete precedent for extracting intrinsic geometry from samples of measures, via graph Dirichlet energies that converge to Laplace-Beltrami-type energies on Wasserstein submanifolds (Ambrosio et al., 2005; Chow & Gangbo, 2019; Oliver et al., 2025; Moosmueller, 2024). In contrast, most policy learning methods, from policy gradient/actor-critic (Sutton et al., 1999; Konda & Tsitsiklis, 1999; Gu et al., 2024) to decision transformers (Chen et al., 2021), update policies primarily through myopic parameter-space gradients or KL-style constraints (Schulman et al., 2017; Achiam et al., 2017), and the abundant historical policies generated during training are rarely treated as samples for explicitly learning or regularizing the induced policy-manifold geometry. In this sense, we learn from experience without truly learning geometry from experience.

In this work, we make the geometry mismatch explicit and treat policy learning as a geometric evolution in Wasserstein space, rather than a sequence of myopic steps in parameter space (Fig. 1). We introduce **Policy Laplacian Trace (PLT)**, an experience-induced policy update principle: at iteration $t$, PLT regards the historical policies $\{\pi_{\theta_i}\}_{i \leq t}$ as samples from the realizable Wasserstein policy manifold, constructs a local OT-induced graph in $\mathcal{P}_2(\mathcal{A})$, and computes the next policy by a variational step that couples a Wasserstein barycentric objective with a KL-proximal term. This yields a task-gradient-driven drift regularized by a history-induced Laplacian smoothing, thereby discouraging geometry-unaware jumps measured by $W_2$ while retaining the local stability benefits of KL-style trust regions. Under standard regularity and sampling conditions, drawing on Laplacian learning on Wasserstein submanifolds, we show that the discrete energy implicitly optimized by PLT consistently approximates a continuum $p$-Dirichlet (and thus $p$-Laplacian) energy on the policy submanifold, providing a measure-geometric justification for the operator. Because PLT only assumes that a policy outputs an action distribution, it applies broadly to policy-based learning in reinforcement learning and sequence modeling. Specifically, our contributions are as follows:

- **Experience-induced Wasserstein policy manifold.** We formalize a measure-geometric view of policy learning: KL/Fisher or parameter-space control need not bound transport-scale change in $W_2$. We model the training trajectory $\{\pi_{\theta_t}(\cdot \mid x)\}_t$ as samples from an experience-induced low-dimensional subset of $\mathcal{P}_2(\mathcal{A})$.

- **Policy Laplacian Trace (PLT).** We propose PLT, a plug-and-play OT+KL proximal update that builds an OT-induced local graph over retrieved historical policies and Laplacian-smooths task-gradient drift, directly calibrating updates in Wasserstein geometry.

- **Theory and evidence.** We show PLT's discrete graph energy consistently approximates a $p$-Dirichlet / Laplace–Beltrami energy on the realizable subset, and demonstrate improved stability, sample efficiency, and forgetting across RL and LLM-as-policy evaluations.

## 2 RELATED WORK

**Laplacian and manifold regularization.**

Laplacian and manifold-regularization methods (e.g., manifold regularization, label propagation, diffusion maps) smooth scalar/vector functions on a data graph by minimizing a graph Dirichlet energy, yielding harmonic extensions or diffusion embeddings that converge to the Laplace–Beltrami operator under dense sampling (Zhu et al., 2003; Belkin et al., 2006; Greilhuber & Kepplinger, 2023; Coifman & Lafon, 2006; Deidda et al., 2026). Nonlinear variants based on graph $p$-Dirichlet energies and $p$-Laplacians further improve boundary sensitivity and robustness in low-label regimes (Slepcev & Thorpe, 2019). These ideas have recently been lifted from Euclidean data to *manifolds of probability measures*. Endowing the action space with a ground metric induces the 2-Wasserstein geometry on $\mathcal{P}_2(\mathcal{A})$, which admits a well-studied Riemannian-like structure with an *Otto calculus* interpretation: tangent vectors can be identified with gradient fields, and Wasserstein gradient flows of suitable functionals correspond to continuity/Fokker–Planck-type PDEs (Ambrosio et al., 2005; Villani et al., 2008; Zhang et al., 2025). Building on this viewpoint, *Laplace learning in Wasserstein*

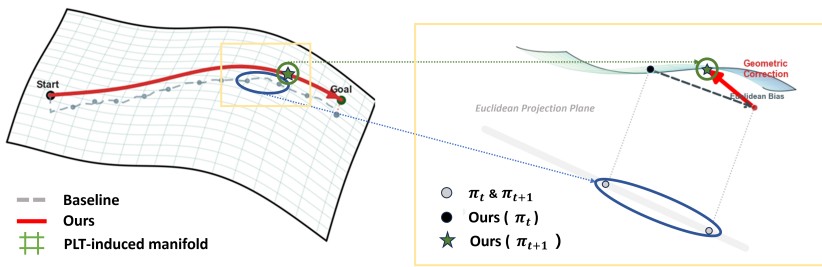

Figure 1: An illustration on how policies change over the course of training in a Wasserstein manifold. Policies lie on an (idealized) low-dimensional Wasserstein policy manifold, with the gray dot denoting the current policy $\pi_t$. A standard update computed in a Euclidean proxy space (parameter/KL–Fisher geometry) follows the dashed direction, introducing an *Euclidean bias* that can drift off the manifold and yield misaligned transport-scale changes. PLT adds an experience-induced geometric correction (red) to realign the step with the manifold, producing $\pi_{t+1}$ (green star). Dashed vertical projections emphasize that small Euclidean/KL-constrained steps may correspond to large $W_2$ displacements. **Left:** top-down view of update directions. **Right:** side view of off-manifold drift and correction.

*space* treats measures as graph nodes, constructs graphs via Wasserstein distances, and establishes $\Gamma$-convergence of discrete $p$-Dirichlet energies to continuous energies on Wasserstein submanifolds with explicit Laplace–Beltrami characterizations (Oliver et al., 2025; Chow & Gangbo, 2019; Moosmueller, 2024). Overall, this line of work remains primarily an offline regularization framework on static graphs, focusing on label or embedding smoothing, whereas our approach treats the policy trajectory as a set of $W_2$ samples, enabling iterative, task-driven decision-making.

**Local Optimization & Wasserstein Geometry in Machine Learning**      Mainstream policy learning (and modern ML training more broadly) largely reduces to local first-order optimization in a Euclidean parameter space: from cross-entropy/maximum-likelihood in supervised and sequence modeling to policy gradient and actor–critic in RL, geometry is introduced only locally via natural gradients, mirror descent, or KL-based trust regions (e.g., TRPO/PPO) that use Fisher information to control inter-iterate drift (Amari, 1998; Kakade, 2001; Peyré et al., 2019; Beck & Teboulle, 2003; Schulman et al., 2017; 2015). This implicitly assumes that small parameter or KL steps reflect the true scale of policy change, yet it does not model policies as conditional measures in $\mathcal{P}_2(\mathcal{A})$ and thus ignores Wasserstein proximity, so KL-controlled updates can still entail large $W_2$ mass transport (notably for multimodal, moving-support, or continuous-action policies) (Bobkov & Götze, 1999; von Renesse & Sturm, 2005; Villani et al., 2008; Xie et al., 2025), yielding oscillations, mode collapse, and out-of-distribution behavior. Meanwhile, OT/Wasserstein has been most influential in generative and diffusion modeling (Cuturi, 2013; De Bortoli et al., 2021), and in RL/IL it is typically used as a distribution-matching loss, reward-shaping signal, or occasional alternative to KL for trust regions (Dadashi et al., 2020; Terpin et al., 2022; Zhang et al., 2018; Salim et al., 2020; He et al., 2022). These uses primarily penalize marginal distribution discrepancies, whereas our method treats the evolving policy sequence as geometric samples and imposes a local Dirichlet/Laplacian trace constraint via a historical $W_2$ graph during training.

# 3   POLICY LEARNING ON EXPERIENCE-INDUCED WASSERSTEIN MANIFOLDS

Following the Preliminaries in Appendix.A, we view policy learning as the evolution of conditional distributions,

$$x \mapsto \pi_\theta(\cdot \mid x) \in \mathcal{P}_2(\mathcal{A}). \tag{1}$$

At iteration $t$, standard policy optimization performs a myopic one-step update

$$\pi_{\theta_{t+1}} = \mathcal{U}_t(\pi_{\theta_t}; \mathcal{T}), \tag{2}$$

where $\mathcal{T}$ denotes a task-dependent operator, such as Bellman evaluation or a policy-gradient objective. Although parameter updates $\theta_{t+1} = \theta_t - \eta \nabla_\theta L_t$ treat each policy $\pi_{\theta_t}$ in isolation, the induced

sequence $\{\pi_{\theta_t}\}_t$ traces a low-dimensional realizable manifold embedded in Wasserstein space. Ignoring this latent geometric structure, purely task-driven updates can induce distributionally jagged policy variations across neighboring states, thereby degrading stability and generalization. We propose to regularize policy learning by its own evolution: each update couples the task objective with Laplacian smoothing in distribution space, enforcing consistency with geometrically nearby historical policies. This leads to the *Policy Laplacian Trace* (PLT) framework, whose components and learning procedure are detailed in the following sections.

## 3.1    POLICY LAPLACIAN TRACE

To incorporate potentially valuable geometric information embedded in the policy manifold into conventional updates, we construct *Policy Laplacian Trace* (PLT).

**Definition 3.1** (Policy Laplacian Trace). At optimisation iteration $t$, we log an evolution sample

$$\mathcal{E}_t = (s_t, \pi_t, \xi_t), \tag{3}$$

where $s_t$ is a visited state (or context), $\pi_t := \pi_{\theta_t}(\cdot \mid s_t) \in \mathcal{P}_2(\mathcal{A})$ is the *conditional policy distribution* at that input, and $\xi_t$ optionally encodes auxiliary local training signals (e.g., a low-dimensional summary of the advantage or gradient direction).

The Policy Laplacian Trace (PLT) at step $t$ is the collection

$$\mathcal{H}_t = \{\mathcal{E}_i\}_{i \leq t}, \tag{4}$$

maintained with a fixed capacity (e.g., FIFO or similarity-based prioritization). Crucially, PLT regularizes in the *output* Wasserstein geometry, not in state space: the input $s$ is used only to retrieve comparable experience (define neighborhoods), while geometry is measured between the retrieved *policy distributions* via $W_2$. Thus $\mathcal{H}_t$ is more than a replay buffer—it provides geometric landmarks in $(\mathcal{P}_2(\mathcal{A}), W_2)$ for policies conditioned on nearby inputs, inducing a local, time-varying geometry for the map $s \mapsto \pi(\cdot \mid s)$ and enabling Laplacian-style regularization in distribution space.

A standard policy optimisation (Eq.2) with PLT is modified through a fusion operator $\mathcal{M}_t$:

$$\pi_{\theta_{t+1}} = \mathcal{M}_t\{\mathcal{U}_t(\pi_{\theta_t}; \mathcal{T}); \ \mathcal{H}_t\}. \tag{5}$$

## 3.2    POLICY LAPLACIAN LEARNING FROM EXPERIENCE

At each iteration, the PLT serves solely as a geometric operator: it is queried to construct a local policy graph, which in turn induces a Wasserstein–Dirichlet energy used to regularize the next update in distribution space. The specific three steps will be presented in following section.

### 3.2.1    POLICY LAPLACIAN CONSTRUCTION

Given the current input/state $s_t$, an $s_t$-centered local experience graph could be constructed. We first retrieve similar past states $s_i$ from PLT $\mathcal{H}_t$ and select the top-$k$ neighbors to form the set $\mathcal{N}_k(s_t)$. The edge weights $w$ in the graph are then obtained by locally normalizing the affinities of these neighbors:

$$w_{tj} = \frac{\kappa(s_t, s_j)}{\sum_{i \in \mathcal{N}_k(s_t)} \kappa(s_t, s_i)}, \qquad j \in \mathcal{N}_k(s_t). \tag{6}$$

where $\kappa$ is a similarity kernel $\kappa : \mathcal{X} \times \mathcal{X} \to \mathbb{R}_+$, which could be instantiated with any task-specific similarity measure, for example:

$$\text{RBF kernel on learned representations:} \kappa(s, s') = \exp\left(-\frac{\|\phi(s) - \phi(s')\|^2}{\sigma^2}\right),$$

$$\text{Attention style affinity: } \kappa(s, s') = \exp\left(\frac{\phi(s)^\top \psi(s')}{\sqrt{d}}\right). \tag{7}$$

The $s_t$-centered graph is sparse, with a topology induced by experience similarity and edge weights that quantify neighbor relevance. By construction, the weights are nonnegative and sum to one (defining a convex combination), so that higher-similarity neighbors receive larger weights. Symmetry depends on the kernel: distance-based kernels typically produce symmetric graphs, whereas attention-based affinities are generally asymmetric, allowing for context-dependent matching.

### 3.2.2 WASSERSTEIN-DIRICHLET FUSION ENERGY

Building on the construction in the previous Section 3.2.1 and Eq.27, to bridge the geometric view with an executable update, we define a local memory potential at iteration $t$ by

$$\Phi_t(\pi_\theta) \coloneqq \frac{1}{2} \sum_{j \in \mathcal{N}(s_t)} w_{tj}\, W_{2,\varepsilon}^2\big(\pi_\theta(\cdot \mid s_t), \pi_\theta(\cdot \mid s_j)\big), \tag{8}$$

Geometrically, $\Phi_t$ acts as a local Wasserstein–Dirichlet energy on the neighborhood graph: its (Wasserstein) first variation induces a graph-Laplacian force in $\mathcal{P}_2(\mathcal{A})$ that pulls the candidate policy toward a weighted barycentric configuration of nearby policies, thereby implementing Laplacian-type smoothing directly in distribution space. Formally, the corresponding Wasserstein subgradient $\nabla_{W_2}\Phi_t(\pi)$ takes the form of a weighted aggregation of logarithmic transport directions from $\pi_\theta$ to its neighbors (Lemma F.5; Remark F.6) and is defined as:

$$\nabla_{W_2}\Phi_t(\pi_\theta) \;\propto\; - \sum_{j \in \mathcal{N}(s_t)} w_{tj}\, \mathrm{Log}_{\pi_\theta}(\pi_\theta(\cdot \mid s_j)), \tag{9}$$

where $\mathrm{Log}_{\pi_\theta}(\pi_\theta(\cdot \mid s_j))$ denotes the (entropic) Wasserstein log direction from $\pi_\theta(\cdot \mid s_t)$ to $\pi_\theta(\cdot \mid s_j)$. We then instantiate the abstract fusion operator $\mathcal{M}_t$ in Eq.5 by solving the proximal objective

$$\pi_{\theta_{t+1}} \;=\; \mathcal{M}_t\{\arg \min_{\pi \in \mathcal{P}_2(\mathcal{A})} \langle \hat{g}_t, \pi_{\theta_t}\rangle \,,\, \lambda\, \Phi_t(\pi_\theta)\}, \tag{10}$$

where $\lambda$ weights the geometry regularizer and $\hat{g}_t$ denotes a task-induced linear functional on $\mathcal{P}_2(\mathcal{A})$, which can be equivalently viewed as the myopic update $\mathcal{U}_t(\pi_\theta)$ in Eq. 2 (e.g., in RL, an empirical estimate of $-A_t(\cdot \mid s_t)$, so that $\langle \hat{g}_t, \pi_{\theta_t}\rangle$ recovers the standard policy-gradient surrogate).

The fusion process in Eq. 10 can be viewed as an implicit (proximal) time-discretization of a composite geometric energy, yielding existence/uniqueness and an energy descent guarantee under mild convexity assumptions (Lemma F.8). See Fig. 1 for a visual intuition of the geometry mismatch and PLT's geometric correction. Appendix.B provides additional details on how to efficiently implement this proximal fusion.

## 4 PLT IMPLEMENTATION

To solve the efficiency and robustness problems, we use PLT as an experience Operator. We consider two canonical realizations of conditional policy learning: Proximal Policy Optimization (PPO) in reinforcement learning, and token-level policy learning in large language models on simple tasks. These settings are selected for clarity rather than generality. PLT is a framework-level construct that naturally generalizes to broader learning paradigms, enabling principled and systematic reuse of experience across domains. Additional implemented algorithm for RL and pseudocodes implementations are provided in Appendix C.1.

### 4.1 LLM INSTANTIATION AND IMPLEMENTATION DETAILS

We view autoregressive LLM training as a policy-learning problem on a discrete action space given by the vocabulary $\mathcal{V}$ (Radford et al., 2018). At training iteration $n$, we do not freeze the model parameters. For a prefix $x_{\leq t} \in \mathcal{V}^t$ at position $t$, with vocabulary $\mathcal{V} = \{1, \ldots, V\}$, the model parameterized by $\theta_n$ outputs a next-token distribution on the probability simplex $\Delta^{V-1} \subset \mathbb{R}^V$:

$$p_{n,t}^{\mathrm{out}} \coloneqq p_{\theta_n}(\cdot \mid x_{\leq t}) \in \Delta^{V-1}, \; h_{n,t} \coloneqq f_{\theta_n}(x_{\leq t}) \in \mathbb{R}^{d_h}, \quad \Delta^{V-1} \coloneqq \Big\{ p \in \mathbb{R}_{\geq 0}^V : \sum_{v=1}^V p(v) = 1 \Big\}, \tag{11}$$

where $h_{n,t}$ denotes the contextual representation of the prefix (e.g., the Transformer hidden state).

To update $\theta_n$, the model is supervised with the ground-truth next token $y_t$ by minimizing the cross-entropy loss $\mathcal{L}_{\mathrm{CE}}$:

$$\mathcal{L}_{\mathrm{CE}}^{(n,t)} = - \log p_{n,t}^{\mathrm{out}}(y_t). \tag{12}$$

### 4.1.1 POLICY LAPLACIAN TRACK IN LLM

To implement Policy Laplacian Track (PLT), we construct position-conditioned memories $\mathcal{M}_\ell$:

$$\mathcal{M}_\ell := \{\mathcal{E}_{n,\ell}\}_{n=1}^{N_{\text{train}}}, \qquad \ell = 1, \ldots, L_{\max}, \tag{13}$$

where $\ell$ indexes the token position and coincides with the current step $t$ when retrieving memories and the PLT $\mathcal{E}_{n,\ell=t}$ is defined as:

$$\mathcal{E}_{n,t} := (h_{n,t}, p_{n,t}^{\text{out}}, s_{n,t}), \tag{14}$$

where $s_{n,t}$ is an uncertainty/difficulty signal such as surprisal $s_{n,t} := -\log p_{n,t}^{\text{out}}(y_t)$.

To fuse PLT and update model given a query at position $\ell = t$ with $(h_{n,t}, p_{n,t}^{\text{out}})$, we retrieve a top-$k$ neighborhood $N_{n,t} \subseteq \mathcal{M}_\ell$ using a similarity score $\kappa(h_{n,t}, h_i)$ (e.g., dot-product or cosine similarity), with softmax weights:

$$w_i(h_{n,t}) = \frac{\exp(\kappa(h_{n,t}, h_i)/\tau_{\text{ret}})}{\sum_{j \in N_{n,t}} \exp(\kappa(h_{n,t}, h_j)/\tau_{\text{ret}})}, \; i \in N_{n,t}, \tag{15}$$

where $\tau_{\text{ret}} > 0$ is a retrieval temperature. The retrieved neighborhood induces a memory distribution:

$$p_{n,t}^{\text{mem}} := \sum_{i \in N_{n,t}} w_i(h_{n,t})\, q_i, \tag{16}$$

where each $q_i \in \Delta^{V-1}$ is the next-token distribution stored in memory.

**Inner view: Fusion with proximal target and gated one-step operator.** Computing the Wasserstein distance $W_2$ over a large vocabulary is computationally expensive due to the $V \times V$ cost matrix. To address this, we adopt a scalable semantic-transport proxy. Let $E \in \mathbb{R}^{V \times d_e}$ be the token embedding matrix with rows $e_a$, $a \in \mathcal{V}$. For any distribution $p \in \Delta^{V-1}$, we define the embedding mean

$$\mu(p) := p^\top E \in \mathbb{R}^{d_e}, \tag{17}$$

and measure the semantic distance between two distributions $p$ and $q$ as

$$d_{\text{sem}}(p, q) := \|\mu(p) - \mu(q)\|_2^2. \tag{18}$$

This reduces the $V \times V$ transport problem to $d_e$ dimensions while preserving the intuition of moving probability mass between semantic clusters.

Therefore PLT defines a proximal target for position $t$ as

$$p_{n,t}^\star \in \arg \min_{q \in \Delta^{V-1}} \left\{ \text{KL}(q \| p_{n,t}^{\text{out}}) + \lambda\, R(q; p_{n,t}^{\text{mem}}) \right\}, \quad R(q; p_{n,t}^{\text{mem}}) = \begin{cases} W_2^2(q, p_{n,t}^{\text{mem}}) \\ \|\mu(q) - \mu(p_{n,t}^{\text{mem}})\|_2^2 \end{cases} \tag{19}$$

Here, the PLT target $p_{n,t}^\star$ can be interpreted as a proximal application of a local memory potential analogous to $\Phi_t$ Eq.8, where $R(q; p_{n,t}^{\text{mem}})$ measures the deviation from the memory distribution, using either the exact Wasserstein distance or its scalable semantic approximation.

Building on this, the fusion is implemented by a single gated step:

$$p_{n,t}^{\text{PLT}} = (1 - g_{n,t})\, p_{n,t}^{\text{out}} + g_{n,t}\, p_{n,t}^{\text{mem}}, \quad g_{n,t} = \sigma\Big( \text{MLP}(h_{n,t}) - \alpha \|\mu(p_{n,t}^{\text{out}}) - \mu(p_{n,t}^{\text{mem}})\|_2^2 \Big), \tag{20}$$

where $\text{MLP}(h_{n,t})$ scores contextual reliability and the semantic displacement term suppresses fusion under large distributional shifts. The fusion is a tractable one-step approximation to $p_{n,t}^\star$, with $g_{n,t}$ controlling interpolation between the model output and memory-fused distribution.

**Outer view: operator-induced targets with hardness gating.** At position $t$, PLT produces the inner-fused distribution $p_{n,t}^{\text{PLT}}$ (e.g., via Eq. 20) and applies it selectively according to token difficulty. With surprisal $s_{n,t} := -\log p_{n,t}^{\text{out}}(y_t)$, define a hardness gate

$$\gamma_{n,t} := \sigma\big(\beta(s_{n,t} - \tau_{\text{hard}})\big) \in [0, 1], \tag{21}$$

where $\tau_{\text{hard}}$ sets the hardness threshold and $\beta$ controls its sharpness, so easy tokens ($\gamma_{n,t} \approx 0$) remain unchanged, while hard tokens ($\gamma_{n,t} \approx 1$) trigger memory and geometry informed fusion. The final operator-induced output is

$$p_{n,t}^{\text{final}} := (1 - \gamma_{n,t}) \, p_{n,t}^{\text{out}} + \gamma_{n,t} \, p_{n,t}^{\text{PLT}} \in \Delta^{V-1}. \tag{22}$$

Optionally, $\gamma_{n,t}$ (or $s_{n,t}$) can be detached to prevent the model from exploiting the gating during training. The parameters are then trained by minimizing the reformed cross-entropy loss:

$$\mathcal{L}_{\text{CE}}^{(n,t),\text{PLT}} = -\log p_{n,t}^{\text{final}}(y_t), \tag{23}$$

## 5 EXPERIMENTS AND EVALUATION

We evaluate the policy Laplacian trace (PLT) in two representative policy-learning settings: reinforcement learning and large language models, using controlled stress tests to isolate failure modes predicted by the geometry-mismatch hypothesis, rather than aiming for broad benchmark SOTA. Ablation on neighborhood size $k$ and compute overhead details are reported in Appendices D.2 and H, respectively.

### 5.1 PLT IN RL

The evaluation is conducted on Atari (Bellemare et al., 2013) and StarCraft II (Samvelyan et al., 2019). Following the Xuance protocol (Liu et al., 2023), all methods within each environment are trained with identical environment interaction budgets, evaluation schedules, and network architectures. We report mean episodic returns for PPO and win rates for MAPPO, averaged over five random seeds (1, 22, 123, 321, 666), with each seed evaluated over multiple episodes without exploration noise. Figure 2 (and in Figure 3 Appendix D) present mean performance curves.

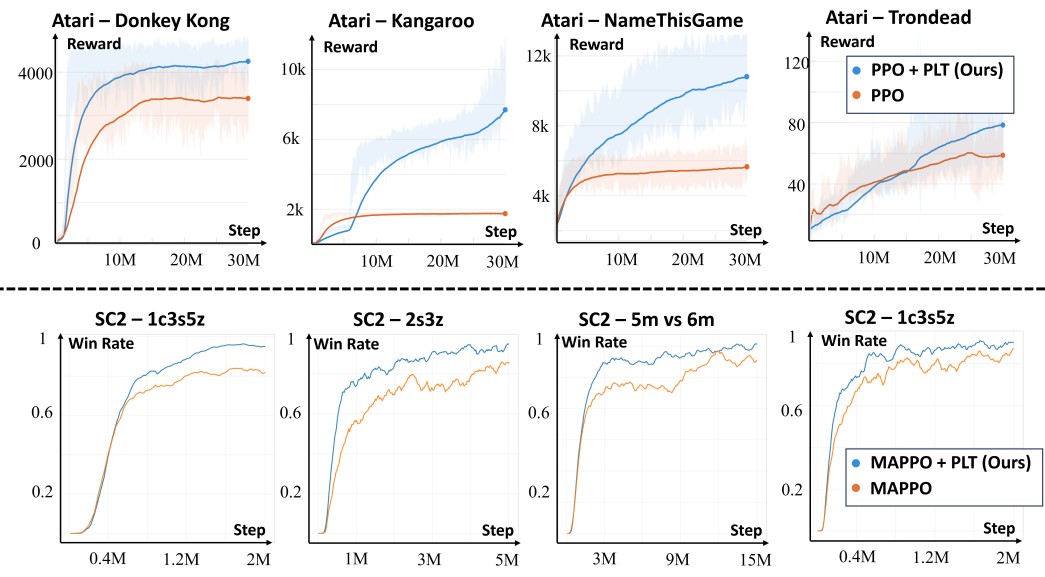

Figure 2: Learning Curves with PLT: PPO Cumulative Returns on Atari (top) and MAPPO Win Rates on StarCraft II (bottom)

**Better Sample Efficiency.** In the Atari game tasks shown in Figure 2, PPO with PLT achieves higher returns at earlier stages of training and reaches strong performance with fewer environment interaction steps, with the effect being most pronounced on Donkey Kong and Name This Game.

**Higher Final Performance.** Figure 2 also shows that RL with PLT consistently converges to higher performance across different environments. In particular, on StarCraft II, MAPPO with PLT

surpasses the baseline convergence bottleneck in all evaluated scenarios. On Atari, although PPO with PLT exhibits lower sample efficiency in the early stages for some games (e.g., Kangaroo and TronDead), it achieves clear performance gains and eventually outperforms the baseline after sufficient training steps.

## 5.2 PLT in LLM

We model *LLMs-as-policies* as conditional distribution learners and evaluate whether PLT's experience-induced Wasserstein regularization improves **stability sample efficiency** beyond scaling via three stress tests: **Counterfactual Trust** (fact–rumor/credibility shifts; *General/Retrieval/Conflict/Official* Acc), **Long-Range Factual Recall** (controlled overwrite at known query–evidence distances; targets-only *Loss (Late)*, *Move/Pass Acc*, *Token Acc (W)*, *NLL (Std)*), and **Few-Shot Learning** (novel-class acquisition with base retention; *Base/New/Overall* Acc). Full dataset construction and protocols are in Appendix G.2. We match budgets/protocols between baselines and +PLT and report multi-seed statistics. Because Table 2 uses strict, conflict-heavy multi-way settings with non-trivial chance levels, we emphasize consistent *lift* and cross-scale trends. Under these controls, PLT consistently improves retrieval-conditioned decisions and robustness, indicating more stable long-range evidence integration beyond scaling (Table 1, Appendix G).

**Counterfactual Trust Task.**    Across scales, +PLT improves retrieval-conditioned decisions under counterfactual and credibility-shifted evidence, with the largest gains typically in *Retrieval* accuracy and frequent improvements in *Conflict/Official* accuracy (Table 2). This benchmark provides a controlled proxy for robustness to evidence shifts and contradictory retrieval.

**Long-Range Factual Recall.**    In a controlled overwrite setting with explicit query–evidence distances and distractors, +PLT consistently reduces *Loss (Late)* and improves far-distance recall while maintaining comparable (or better) calibration (*NLL (Std)*; Table 3). These results isolate distance-driven degradation and are consistent with more stable long-range evidence integration.

**Few-Shot Learning.**    Under base-class retention plus novel-class acquisition, +PLT increases *New* accuracy while keeping *Base* accuracy essentially unchanged (Table 4). This indicates improved sample efficiency for few-shot acquisition with reduced forgetting, complementing the retrieval- and long-context stress tests.

## 6 Conclusion

We study a geometry mismatch in mainstream policy learning: parameter steps or KL trust regions need not reflect transport-scale changes measured by the Wasserstein distance, which can yield irregular distributional trajectories under long-range evidence or distribution shift. We propose Policy Laplacian Trace (PLT), a plug-and-play regularizer that treats historical policies as experience-induced samples of a locally low-dimensional subset in Wasserstein space and anchors updates to neighborhood distributions via Wasserstein potentials (see Appendix E for overhead). Across reinforcement learning and language-model-as-policy stress tests, PLT consistently improves training reliability and cross-distribution performance, supporting Wasserstein-structured history as a practical way to regulate effective update scale. Future work includes extending PLT to larger-scale backbones and broader benchmarks with stronger stability diagnostics.

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

## A   Preliminaries

This section collects the optimal-transport preliminaries used throughout. We treat a policy as a conditional measure $\pi_\theta(\cdot \mid x) \in \mathcal{P}_2(A)$ and compare policies via the 2-Wasserstein geometry.

**Hypothesis 1** (Experience-induced Wasserstein policy geometry is learnable and behavior-relevant). *Fix $x$ and equip $(\mathcal{A}, d_\mathcal{A})$ with a task-semantic metric; let $\Lambda_x := \{\pi_\theta(\cdot \mid x) : \theta \in \Theta\} \subset \mathcal{P}_2(\mathcal{A})$ and denote by $\widetilde{\Lambda}_x \subset \Lambda_x$ the set of policies visited during training. Assume $\widetilde{\Lambda}_x$ lies on a locally $d_x$-dimensional ($d_x \ll \dim \Theta$) bi-Lipschitz chart under $(\|\cdot\|_2, W_2)$ and is sampled densely enough along $\{\pi_{\theta_t}(\cdot \mid x)\}_t$ to consistently estimate $W_2$-kNN neighborhoods and graph-Laplacians from historical policies. Moreover, rewards are Lipschitz in action, $|r(x, a) - r(x, a')| \le L_r d_\mathcal{A}(a, a')$ (optionally $W_1(P(\cdot \mid x, a), P(\cdot \mid x, a')) \le L_P d_\mathcal{A}(a, a')$), so for any $\pi, \pi'$,*

$$\left| \mathbb{E}_{a \sim \pi(\cdot|x)}[r(x, a)] - \mathbb{E}_{a \sim \pi'(\cdot|x)}[r(x, a)] \right| \le L_r \, W_2\big(\pi(\cdot \mid x), \pi'(\cdot \mid x)\big). \tag{24}$$

Appendix D.3 reports diagnostics of low intrinsic dimension/spectral decay in policy snapshots and shows that +PLT reduces $W_2$ drift at comparable KL, supporting the geometry-mismatch motivation and Hypothesis 1.

Let $(A, d_A)$ be a Polish metric space, where $A \subset \mathbb{R}^m$ or a finite action set equipped with a cost matrix. We write $\mathcal{P}_2(A)$ for the set of probability measures on $A$ with finite second moment, and $\mathcal{B}(A)$ for the Borel $\sigma$-algebra. Given a measurable map $T : A \to A$ and a measure $\mu \in \mathcal{P}_2(A)$, the pushforward measure $T_\# \mu$ is defined by

$$\int_A \varphi(a) \, \mathrm{d}(T_\# \mu)(a) = \int_A \varphi(T(a)) \, \mathrm{d}\mu(a), \quad \forall \varphi \in L^\infty(A). \tag{25}$$

For two measures $\mu, \nu \in \mathcal{P}_2(A)$, let $\Pi(\mu, \nu)$ denote the set of couplings on $A \times A$ with marginals $(\mu, \nu)$. The 2-Wasserstein distance is

$$W_2^2(\mu, \nu) := \inf_{\pi \in \Pi(\mu,\nu)} \int_{A \times A} d_A(a, a')^2 \, \mathrm{d}\pi(a, a'). \tag{26}$$

In computation, we use an entropic-regularized variant:

$$W_{2,\varepsilon}^2(\mu, \nu) := \min_{\pi \in \Pi(\mu,\nu)} \int d_A(a, a')^2 \, \mathrm{d}\pi(a, a') + \varepsilon \, \mathrm{KL}(\pi \,\|\, \mu \otimes \nu), \tag{27}$$

which admits efficient Sinkhorn-type solvers in discrete settings and stable sample-based approximations for $\varepsilon > 0$.

Finally, any measures $\{\mu_i\}_{i=1}^k \subset \mathcal{P}_2(A)$ and weights $w \in \Delta^{k-1}$, an (entropic) Wasserstein barycenter is defined by

$$\mathrm{Bar}_\varepsilon(\{\mu_i, w_i\}) \in \arg \min_{\mu \in \mathcal{P}_2(A)} \sum_{i=1}^k w_i \, W_{2,\varepsilon}^2(\mu, \mu_i). \tag{28}$$

## B   OT-Fuse: Efficient Approximate Solver for Proximal Fusion

Based on the Policy Laplacian learning concepts introduced in Section 4, we briefly present OT-FUSE, a lightweight routine that approximately solves this variational update using a small number of Sinkhorn iterations. After turns the patch into a Wasserstein–Dirichlet potential $\Phi_t(\pi)$ that penalizes geometric roughness relative to nearby historical policies. Then updates the current policy by solving the proximal fusion objective (Eq. 10), which couples the task signal with Laplacian-style smoothing in distribution space.

Directly minimizing Eq. 10 can be expensive because it combines a task functional, multiple Wasserstein terms, and a trust-region constraint in policy space. We therefore adopt a two-step approximation: first form a *task-leading* policy $\tilde{\pi}_t$ by applying the base (myopic) update,

$$\tilde{\pi}_t = \mathcal{U}_t(\pi_t; \mathcal{T}), \tag{29}$$

and then fuse $\tilde{\pi}_t$ with the local geometric prior induced by $\mathcal{H}_t$. Concretely, we approximate Eq. 10 by the anchored barycentric proximal problem

$$\pi_{t+1} \;=\; \arg\min_{\pi \in \mathcal{P}_2(\mathcal{A})} \lambda\, \Phi_t(\pi) \;+\; \frac{1}{\eta}\, \mathrm{KL}(\pi\|\tilde{\pi}_t), \tag{30}$$

where $\Phi_t(\pi) = \frac{1}{2} \sum_{j \in \mathcal{N}_k(s_t)} w_{tj} W_{2,\varepsilon}^2(\pi, \pi_j)$. Intuitively, Eq. equation 30 moves the task-leading policy toward a local Wasserstein barycentric configuration of historical neighbors, while the KL term keeps the step conservative.

The resulting OT-Fuse iteration admits a geometric optimization interpretation: it implements an explicit Euler step of the entropic Wasserstein gradient flow induced by $\Phi_t$, and provides a first-order approximation to the corresponding Wasserstein proximal point (Proposition F.11). Assume $\mathcal{A} = \{a_1, \ldots, a_n\}$ with ground cost matrix $C \in \mathbb{R}_+^{n \times n}$, e.g., $C_{uv} = d_{\mathcal{A}}(a_u, a_v)^2$. Let $K := \exp(-C/\varepsilon)$ be the entropic OT kernel. For entropic OT, a coupling between a candidate $\pi$ and a neighbor $\pi_j$ takes the Sinkhorn form $P_j = \mathrm{diag}(u_j)\, K\, \mathrm{diag}(v_j)$ with marginals $P_j \mathbf{1} = \pi$ and $P_j^\top \mathbf{1} = \pi_j$. The anchored barycenter equation 30 can be approximated by a few iterations of a Sinkhorn-barycenter fixed point, warm-started from $\tilde{\pi}_t$.

---

**Algorithm 1** OT-FUSE: few-step entropic barycenter with KL anchor (discrete actions)

---

**Require:** Anchor policy $\tilde{\pi} \in \Delta^n$, neighbors $\{\pi_j\}_{j=1}^k \subset \Delta^n$, weights $w_j \geq 0$ ($\sum_j w_j = 1$), cost $C$, entropic $\varepsilon$, inner steps $L$, anchor exponent $\beta \in (0, 1)$.
**Ensure:** Fused policy $\pi_{\mathrm{fuse}} \in \Delta^n$
  1: $K \leftarrow \exp(-C/\varepsilon)$; $\pi^{(0)} \leftarrow \tilde{\pi}$                                         // Sinkhorn kernel + init
  2: **for** $\ell = 0, 1, \ldots, L - 1$ **do**
  3:     **for** $j = 1, \ldots, k$ **do**
  4:         $v_j \leftarrow \pi_j \oslash (K^\top u_j)$;   $u_j \leftarrow \pi^{(\ell)} \oslash (K v_j)$
  5:         $m_j \leftarrow K v_j$                                        // message to barycenter
  6:     **end for**
  7:     $\pi^{(\ell+1)} \leftarrow \mathrm{Normalize}\!\left(\tilde{\pi}^\beta \odot \prod_{j=1}^k m_j^{(1-\beta)w_j}\right)$
  8: **end for**
  9: Return $\pi_{\mathrm{fuse}} \leftarrow \pi^{(L)}$

---

Here $\odot$ and exponents are taken elementwise, $\oslash$ denotes elementwise division, and $\mathrm{Normalize}(\cdot)$ rescales a positive vector to sum to one. The parameter $\beta$ controls the strength of the KL anchor: larger $\beta$ keeps $\pi_{t+1}$ closer to $\tilde{\pi}_t$, while smaller $\beta$ increases geometric fusion. In practice, $L$ can be very small (e.g., $L \in \{1, 2, 3\}$) with warm-started scalings, making OT-FUSE a lightweight drop-in solver.

OT-FUSE can be understood as a fast approximate proximal step toward a local Wasserstein barycenter induced by the neighborhood $\{\pi_j\}$. Viewed through the minimizing-movement perspective, Eq. 30 is an implicit (proximal) time discretization of a drift–diffusion dynamics on the policy manifold: the task step produces $\tilde{\pi}_t$ (drift), while the Dirichlet potential $\Phi_t$ induces Laplacian-style smoothing (diffusion) in $W_{2,\varepsilon}$ geometry. OT-FUSE implements this step approximately using entropic OT and a few Sinkhorn iterations.

---

**Algorithm 2** Policy Laplacian Trace (PLT): task update + Wasserstein Laplacian fusion

---

**Require:** Stepsize $\eta$, geometry weight $\lambda$, mixed trust weight $\gamma$, entropic OT $\varepsilon$, capacity $M$, $k$NN size $k$, similarity kernel $\kappa$, initial params $\theta_0$.

1: $\mathcal{H} \leftarrow \emptyset$; initialize policy $\pi_{\theta_0}$
2: **for** $t = 0, 1, 2, \ldots$ **do**
3:     Observe context/state $s_t$; current policy $\pi_t \leftarrow \pi_{\theta_t}(\cdot \mid s_t)$
4:     Log trace item $\mathcal{E}_t \leftarrow (s_t, \pi_t, \xi_t)$ *(optional $\xi_t$: advantage/grad summary)*
5:     $\mathcal{H} \leftarrow \textsc{Insert}(\mathcal{H}, \mathcal{E}_t; M)$ *(FIFO / similarity-prune)*
6:     Retrieve neighbors $\mathcal{N} \leftarrow \textsc{kNN}(s_t, \mathcal{H}, \kappa, k)$; weights $w_j \propto \kappa(s_t, s_j)$ for $j \in \mathcal{N}$ (normalize to $\sum_j w_j = 1$)
7:     Task-leading update (any base learner): $\tilde{\theta}_t \leftarrow \textsc{BaseUpdate}(\theta_t)$; $\tilde{\pi}_t \leftarrow \pi_{\tilde{\theta}_t}(\cdot \mid s_t)$
8:     Fuse in distribution space:

$$\pi_{t+1} \leftarrow \begin{cases} \textsc{OT-Fuse}(\tilde{\pi}_t, \{\pi_j\}_{j \in \mathcal{N}}, \{w_j\}, \lambda, \varepsilon) & \mathcal{N} \neq \emptyset \\ \tilde{\pi}_t & \mathcal{N} = \emptyset \end{cases} \tag{31}$$

9:     Project back to parametric policy (distillation / projection):

$$\theta_{t+1} \leftarrow \arg\min_{\theta} \ D(\pi_{t+1} \,\|\, \pi_\theta(\cdot \mid s_t)), \quad D = D_{\mathrm{KL}} + \frac{\gamma}{2} W_{2,\varepsilon}^2 \tag{32}$$

10: **end for**

---

## C Applied PLT Algorithm

### C.1 RL/MDP Instantiation and Implementation Details

We implement PLT Eq. 10 in an *amortized* manner: for each state $s$ sampled from the rollout buffer, we compute a conservative reference policy $\pi_{\mathrm{align}}(\cdot \mid s)$ as a tractable approximation to the local proximal solution, and use it as an anchor in the PPO surrogate objective (Sutton et al., 1998; Schulman et al., 2017). Given a MDP with PPO algorithm, an on-policy rollout buffer $\mathcal{B}$ of horizon $T$ under the behavior policy $\pi_{\theta_{\mathrm{old}}}$ is defined as

$$\mathcal{B} = \left\{ (s_t, a_t, r_t, s_{t+1}, \log \pi_{\theta_{\mathrm{old}}}(a_t \mid s_t), \widehat{A}_t) \right\}_{t=1}^T, \tag{33}$$

where $s$ is state, $a$ is action, $r$ is reward and $\widehat{A}_t$ is advantage. The actor objective satisfies

$$\mathcal{L}_{\mathrm{actor}}(\theta) = \mathbb{E}_{(s,a,\hat{A}) \sim \mathcal{B}} \left[ \min\left( \frac{\pi_\theta(a_i|s_i)}{\pi_{\theta_{\mathrm{old}}}(a_i|s_i)} \, \hat{A}, \mathrm{clip}\left( \frac{\pi_\theta(a_i|s_i)}{\pi_{\theta_{\mathrm{old}}}(a_i|s_i)}, 1 - \epsilon, 1 + \epsilon \right) \hat{A} \right) \right], \tag{34}$$

In parallel, we maintain a fixed-capacity Policy Laplacian Trace $\mathcal{H}$ that logs state-indexed local action distributions at each time step $t$,

$$\mathcal{H} \leftarrow \mathcal{H} \cup \{(z_t, \varphi_t)\}, \ z_t = f_{\mathrm{enc}}(s_t), \ \varphi_t := \pi_{\theta_{\mathrm{old}}}(\cdot \mid s_t), \tag{35}$$

where $z_t$ is a retrieval key (state embedding) and $\varphi_t$ is a lightweight representation of the policy distribution at $s_t$. For continuous actions, we store diagonal-Gaussian parameters $\varphi_t = (\mu_t, \log \sigma_t)$; for discrete actions, we store categorical probabilities or logits $\varphi_t = p_t$.

Given a query state $s$ (e.g., from PPO minibatches), we compute its key $z = f_{\mathrm{enc}}(s)$ using the same encoder as the policy (e.g., the penultimate feature layer), and retrieve the top-$k$ trace indices $I(s) \subset \{1, \ldots, |\mathcal{H}|\}$ under Euclidean or cosine distance in key space. Concretely, the Eq.6 is realized by a temperature-controlled normalized kernel for computing the retrieval weights $w$,

$$w_i(s) = \frac{\exp(-\|z - z_i\|^2 / \tau)}{\sum_{j \in I(s)} \exp(-\|z - z_j\|^2 / \tau)}, \quad \sum_{i \in I(s)} w_i(s) = 1, \tag{36}$$

where $\tau$ is a temperature. Given neighbors $I(s)$ and weights $\{w_i(s)\}$, we form a state-conditioned historical reference $\pi_{\mathrm{hist}}(\cdot \mid s)$ with $\varphi \in H$: for discrete actions,

$$\pi_{\mathrm{hist}}(\cdot \mid s) \leftarrow \sum_{i \in I(s)} w_i(s) \, p_i, \tag{37}$$

while for continuous actions we use a weighted Gaussian mixture

$$\pi_{\text{hist}}(\cdot \mid s) \leftarrow \sum_{i \in I(s)} w_i(s)\mathcal{N}(\mu_i, \sigma_i^2). \tag{38}$$

We then compute a conservative Wasserstein alignment as a computable KL-regularized proximal application of the local memory potential $\Phi_t$ Eq.8 by the objective

$$\pi_{\text{align}}(\cdot \mid s) \in \arg\min_{\pi \in \mathcal{P}_2(\mathcal{A})} \left\{ W_{2,\varepsilon}^2\big(\pi, \pi_{\text{hist}}(\cdot \mid s)\big) + \rho\,\text{KL}\big(\pi \| \pi_\theta(\cdot \mid s)\big) \right\}, \tag{39}$$

approximated via sample-based entropic OT: draw $A = \{a_i\}_{i=1}^m \sim \pi_\theta(\cdot \mid s)$ and $B = \{b_j\}_{j=1}^m \sim \pi_{\text{hist}}(\cdot \mid s)$, set $C_{ij} = \|a_i - b_j\|_2^2$, compute the Sinkhorn plan

$$P^\star \in \arg\min_{P \in \mathcal{U}(u,v)} \langle C, P \rangle + \varepsilon H(P), \qquad u = v = \tfrac{1}{m}\mathbf{1}, \tag{40}$$

and take barycentric projections $\tilde{a}_i = \sum_j \big(P_{ij}^\star / \sum_{j'} P_{ij'}^\star\big) b_j$ followed by a conservative step

$$a_i^{\text{align}} = (1-\alpha)a_i + \alpha\tilde{a}_i, \quad \alpha = \alpha(\rho) \ \ (\text{e.g. } \alpha = \tfrac{1}{1+\rho}), \tag{41}$$

after which $\pi_{\text{align}}(\cdot \mid s)$ is obtained by refitting the policy family to $\{a_i^{\text{align}}\}_{i=1}^m$ (moment matching for Gaussians; direct entropic OT on categorical support for discrete actions).

We incorporate $\pi_{\text{align}}$ into PPO as a conservative, geometry-aware anchor by optimizing the unified actor objective

$$\mathcal{L}_{\text{actor}}(\theta) = \mathbb{E}_{(s,a,\hat{A}) \sim \mathcal{B}}\left[ \min\left( \tfrac{\pi_{align}(a_i \mid s_i)}{\pi_{\theta_{\text{old}}}(a_i \mid s_i)}\,\hat{A}, \text{clip}\big( \tfrac{\pi_{align}(a_i \mid s_i)}{\pi_{\theta_{\text{old}}}(a_i \mid s_i)}, 1-\epsilon, 1+\epsilon \big)\,\hat{A} \right) \right], \tag{42}$$

In all cases, $\pi_{\text{align}}$ is computed from $\pi_\theta$ in Eq.34 and the PLT $\mathcal{H}$ and is held fixed within each PPO update epoch, consistent with PPO's fixed-behavior-policy protocol.

To amortize OT cost, we precompute and cache $\pi_{\text{align}}(\cdot \mid s)$ for all states in the rollout buffer $\mathcal{B}$ once per PPO iteration (or only for a subset of high-importance states, e.g., the top-quantile of $|\hat{A}|$). When the trace is not yet sufficiently populated (early training) or retrieval is unreliable, we fall back to the identity anchor $\pi_{\text{align}} = \pi_{\theta_{\text{old}}}$. Per queried state, the dominant overhead is Sinkhorn on an $m \times m$ cost matrix, i.e., $\mathcal{O}(m^2 I)$ for $I$ Sinkhorn iterations, with small $m$ (e.g., 8–32) sufficient in practice. Trace storage scales as $\mathcal{O}\big(M(d_z + d_\varphi)\big)$, where $d_z$ is the key dimension and $d_\varphi$ is the distribution-parameter dimension (Gaussian: $d_\varphi = 2d_a$; categorical: $d_\varphi = |\mathcal{A}|$).

## C.2 PSEUDOCODE: PPO + PLT

---

**Algorithm 3** PPO with Policy Laplacian Trace

---

1: **Hyper:** clip $\epsilon$, neighbors $k$, batch size $b$
2: Initialize policy $\pi_\theta$, PLT memory $\mathcal{H}$ (capacity $M$), Reply Buffer $\mathcal{B}$ (capacity $T$)
3: **for** step $t = 1, 2, \ldots$ **do**
4:     Observe $s_t$, take $a_t \sim \pi_\theta(\cdot|s_t)$, receive $(r_t, s_{t+1})$; store $(s_t, a_t, r_t, s_{t+1}, \log \pi_{\theta_{\text{old}}}(a_t|s_t), \hat{A}_t)$ in $\mathcal{B}$
5:     **if** step $t \geq$ start learning step **then**
6:         Sampling $(s_i, a_i, r_i, s_{i+1}, \log \pi_{\theta_{\text{old}}}(a_i|s_i), \hat{A}_i)_{i=1}^b$ from $\mathcal{B}$
7:         **if** $|\mathcal{H}| < k$ **then**
8:             $\pi_{\text{align}}(\cdot|s) \leftarrow \pi_{\theta_{\text{old}}}(\cdot|s)$ {fallback}
9:             **continue**
10:         **else**
11:             Retrieve $I(s_i) = $ top-$k$ neighbors in $\mathcal{H}$ and compute $w_i(s_i)$ (cf.36)
12:             **if** $\pi_\theta$ is discrete (categorical) **then**
13:                 $\pi_{\text{hist}} \leftarrow \sum_{i \in I(s_i)} w_i \, p_i$ (cf.37)
14:             **else if** $\pi_\theta$ is continuous (gaussian) **then**
15:                 $\pi_{\text{hist}} \leftarrow \sum_{i \in I(s_i)} w_i \, \mathcal{N}(\mu_i, \sigma_i^2)$ (cf.38)
16:                 Estimate $\pi_{\text{align}}$ through Sinkhorn plan and barycentric projection (cf.39 $\sim$ cf.41)
17:             **end if**
18:         **end if**
19:         **Policy Update (cf.42):**

$$\mathbb{E}_{(s,a,\hat{A}) \sim \mathcal{B}} \left[ \min \left( \frac{\pi_{\text{align}}(a|s)}{\pi_{\theta_{\text{old}}}(a|s)} \hat{A}, \ \text{clip}(\cdot, 1 \pm \epsilon) \hat{A} \right) \right]$$

20:         Update $\theta$ and Push $(s_t, \varphi_t := \pi_{\theta_{\text{old}}}(\cdot|s_t))$ into PLT $\mathcal{H}$
21:     **end if**
22: **end for**

---

## C.3 PSEUDOCODE: LLM + PLT

---

**Algorithm 4** LLM Training with PLT

---

1: **Input:** dataset $\mathcal{D}$, parameters $\theta_0$, memory $\mathcal{M} \leftarrow \emptyset$, min-memory $M_{\min}$, $k$NN size $k$, hardness threshold $\tau$, gates $\alpha, \beta$, step size $\eta$.
2: **for** $t = 0, 1, \ldots, T$ **do**
3:      Sample a minibatch $\{(x_b, y_b)\}_{b=1}^{B} \sim \mathcal{D}$.
4:      Compute logits $z_b = f_\theta(x_b)$ and output distributions $p_b^{\mathrm{out}} = \mathrm{Softmax}(z_b)$.
5:      **for** $b = 1, \ldots, B$ **do**
6:          **if** $|\mathcal{M}| \geq M_{\min}$ **then**
7:              Retrieve $\mathcal{N}_k(x_b) = \{(h_i, p_i)\}$ by $k$NN over keys $(h_b)$.
8:              Weights $w_i = \mathrm{Softmax}(\kappa(h_b, h_i))$ and memory barycenter $p_b^{\mathrm{mem}} = \sum_{i \in \mathcal{N}_k} w_i \, \mathrm{stopgrad}(p_i)$.
9:              Fusion gate $g_b^{\mathrm{fuse}} = \sigma\Big(\mathrm{MLP}(h_b) - \alpha \, d_{\mathrm{OT}}(p_b^{\mathrm{out}}, p_b^{\mathrm{mem}})\Big)$.
10:             Fused distribution (no gradient): $p_b^{\mathrm{fuse}} = \mathrm{stopgrad}\big((1 - g_b^{\mathrm{fuse}})p_b^{\mathrm{out}} + g_b^{\mathrm{fuse}} p_b^{\mathrm{mem}}\big)$.
11:         **else**
12:             $p_b^{\mathrm{fuse}} \leftarrow \mathrm{stopgrad}(p_b^{\mathrm{out}})$.
13:         **end if**
14:     **end for**
15:     Surprisal $s_b = -\log p_b^{\mathrm{out}}(y_b)$ and hardness gate $\gamma_b = \sigma(\beta(s_b - \tau))$.
16:     Geometry-aware target $\pi_b^{\mathrm{target}} = (1 - \gamma_b) \, \mathrm{stopgrad}(p_b^{\mathrm{out}}) + \gamma_b \, p_b^{\mathrm{fuse}}$.
17: **end for**
18: Optimize with supervised CE plus distributional regularization:

$$\mathcal{L}(\theta) = \frac{1}{B} \sum_{b=1}^{B} \Big( \underbrace{-\log p_b^{\mathrm{out}}(y_b)}_{\text{CE to labels}} + \lambda \underbrace{H(\pi_b^{\mathrm{target}}, p_b^{\mathrm{out}})}_{\text{PLT distillation}} \Big),$$

19: $\theta_{t+1} \leftarrow \theta_t - \eta \nabla_\theta \mathcal{L}(\theta)$.
20: Log experience: $\mathcal{M} \leftarrow \mathcal{M} \cup \{(\mathrm{stopgrad}(h_b), \mathrm{stopgrad}(p_b^{\mathrm{out}}))\}_{b=1}^{B}$.

---

# D    MORE EXPERIMENT

## D.1    MORE EXPERIMENT RESULTS

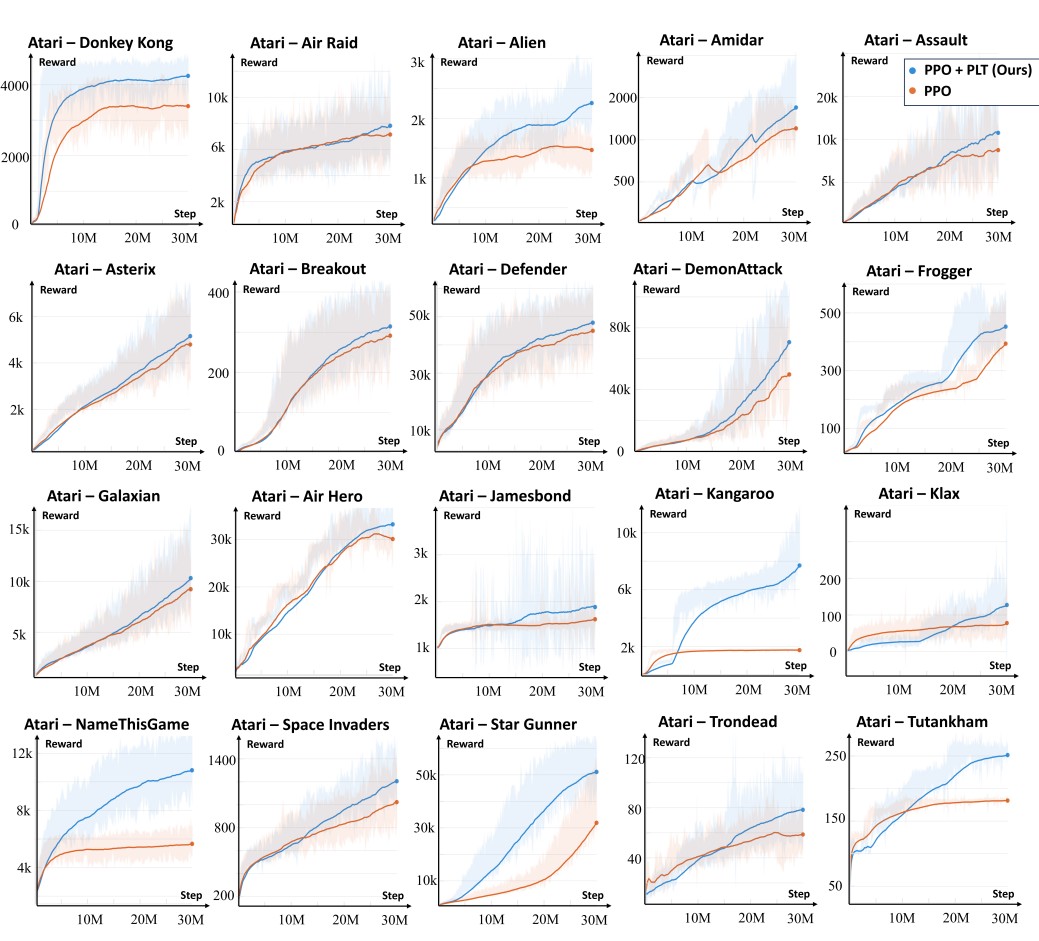

Figure 3: Learning curves with PLT compare with PPO baseline cumulative returns on Atari Games.

Table 1: **PLT improves robustness across three settings.** We report Base and +PLT for each metric. Green indicates improvement over Base: positive relative gains for accuracies, and negative relative drops for losses. Full breakdowns are in Tables 2, 3, and 4.

| Model | Param. | (I) Multi-Category Accuracy | | | | (II) Long-Range Factual Recall | | | | (III) Few-Shot New Categories | | | |
| | | General Acc ↑ | | Retrieval Acc ↑ | | Loss (Late) ↓ | | Token Acc (W) ↑ | | New Acc ↑ | | Overall ↑ | |
| | | Base | +PLT | Base | +PLT | Base | +PLT | Base | +PLT | Base | +PLT | Base | +PLT |
| GPT-Nano | 2.5M | 8.38% | **9.09%**(+8.5%) | 8.38% | **10.62%**(+26.7%) | 1.58 | **1.01**(-36.1%) | 56.9% | **59.6%**(+4.7%) | 0.00% | 0.00% | 70.00% | 70.00% |
| GPT-Micro | 5M | 6.11% | **10.39%**(+70.0%) | 7.02% | **11.20%**(+59.5%) | 2.03 | **1.20**(-40.9%) | 48.8% | **51.8%**(+6.1%) | 10.78% | **39.22%**(+263.8%) | 78.00% | **85.00%**(+9.0%) |
| GPT-Mini | 10M | 8.35% | **10.15%**(+21.6%) | 9.71% | **11.49%**(+18.3%) | 1.35 | **1.05**(-22.2%) | 58.7% | **59.3%**(+1.0%) | 13.86% | **16.83%**(+21.4%) | 82.00% | **83.00%**(+1.2%) |
| Gopher | 44M | 6.05% | **11.00%**(+81.8%) | 7.10% | **12.50%**(+76.1%) | 1.84 | **1.16**(-37.0%) | 54.5% | **58.0%**(+6.4%) | 0.97% | **13.59%**(+1301.0%) | 85.00% | **90.00%**(+5.9%) |
| OpenAI-GPT | 117M | 9.23% | **10.22%**(+10.7%) | 10.53% | **11.78%**(+11.9%) | 1.80 | **1.68**(-6.7%) | 58.7% | **60.0%**(+2.2%) | 0.00% | **0.99%**(+∞%) | 88.00% | **93.00%**(+5.7%) |
| GPT-2 | 124M | 10.83% | **12.20%**(+12.7%) | 12.32% | **13.77%**(+11.8%) | 1.71 | **1.01**(-40.9%) | 56.3% | **59.7%**(+6.0%) | 2.13% | **14.89%**(+599.1%) | 89.00% | **90.00%**(+1.1%) |

Table 2: **Comprehensive Performance Analysis.** "+ PLT." denotes our Wasserstein method. **Bold** indicates the best performance in each category. Green values represent significant gains over the baseline, and shaded cells highlight key zero-to-one breakthroughs.

| Model | Param. | General Acc. | | Retrieval Acc. | | Official Acc. | | Conflict Acc. | |
|---|---|---|---|---|---|---|---|---|---|
| | | Base | + PLT. | Base | + PLT. | Base | + PLT. | Base | + PLT. |
| GPT-Nano | 2.5M | 8.38% | **9.09%** (+8.5%) | 8.38% | **10.62%** (+26.7%) | 0.0% | **20.0%** (+∞%) | 10.53% | 10.53% |
| GPT-Micro | 5M | 6.11% | **10.39%** (+70.0%) | 7.02% | **11.20%** (+59.5%) | 0.0% | 0.0% | 6.98% | **11.26%** (+61.3%) |
| GPT-Mini | 10M | 8.35% | **10.15%** (+21.6%) | 9.71% | **11.49%** (+18.3%) | 0.0% | 0.0% | 10.00% | **11.22%** (+12.2%) |
| Gopher | 44M | 6.05% | **11.00%** (+81.8%) | 7.10% | **12.50%** (+76.1%) | 0.0% | 0.0% | 7.28% | **12.23%** (+68.0%) |
| OpenAI-GPT | 117M | 9.23% | **10.22%** (+10.7%) | 10.53% | **11.78%** (+11.9%) | 0.0% | **12.5%** (+∞%) | 10.61% | **12.86%** (+21.2%) |
| GPT2 | 124M | 10.83% | **12.20%** (+12.7%) | 12.32% | **13.77%** (+11.8%) | 0.0% | 0.0% | 8.79% | **13.11%** (+49.1%) |
| GPT2-Medium | 355M | 11.28% | 11.28% | 12.00% | **12.65%** (+5.4%) | 0.0% | **25.0%** (+∞%) | 11.89% | 10.64% |
| GPT2-Large | 774M | 11.26% | **15.82%** (+40.5%) | 10.00% | **18.18%** (+81.8%) | 40.0% | **40.0%** | 8.33% | **17.06%** (+104.8%) |

Table 3: **Training & Long-Context Bucket Performance.** "+PLT" denotes our Policy Laplacian Trace variant. **Bold** indicates the better result within each metric (higher is better for accuracies; lower is better for loss/NLL). Green denotes improvements over baseline.

| Model | Loss (Late) ↓ | | Move Acc ↑ | | Pass. Acc ↑ | | Token Acc (W) ↑ | | NLL ↓ (Std) | |
|---|---|---|---|---|---|---|---|---|---|---|
| | Base | +PLT | Base | +PLT | Base | +PLT | Base | +PLT | Base | +PLT |
| Gopher | 1.84 | **1.16** | 45.6% | **46.6%** | 32.4% | **34.0%** | 54.5% | **58.0%** | 2.87 (—) | **2.81 (1.83)** |
| GPT-Nano | 1.58 | **1.01** | 34.1% | **34.6%** | 41.8% | **42.7%** | 56.9% | **59.6%** | 2.96 (—) | **2.81 (1.90)** |
| GPT-Micro | 2.03 | **1.20** | 48.8% | **49.3%** | 13.2% | **22.5%** | 48.8% | **51.8%** | 3.19 (—) | **2.97 (1.82)** |
| GPT-Mini | 1.35 | **1.05** | 46.1% | **47.7%** | 35.7% | **40.5%** | 58.7% | **59.3%** | 2.56 (—) | **2.44 (1.68)** |
| OpenAI GPT | 1.80 | **1.68** | 45.2% | **47.9%** | 33.8% | **40.7%** | 58.7% | **60.0%** | 2.80 (—) | **2.71 (2.00)** |
| GPT-2 | 1.71 | **1.01** | 41.8% | **43.9%** | 30.4% | **32.3%** | 56.3% | **59.7%** | 2.92 (—) | **2.58 (1.86)** |
| GPT-2 Medium | 1.70 | **1.23** | 40.6% | **42.5%** | 37.8% | **42.8%** | 54.6% | **55.8%** | 2.90 (—) | **2.58 (1.86)** |

Table 4: **Policy Laplacian Trace (PLT) Enhanced Few-Shot Learning. +PLT** denotes our Wasserstein memory method with Policy Laplacian Trace. **Bold** indicates the best performance within each row. Green values denote absolute improvements over baseline, green shading highlights significant few-shot gains.

| Model | Param. | Base Categories | | Few-Shot (New Cat.) | | Overall | | Few-Shot Gain | |
|---|---|---|---|---|---|---|---|---|---|
| | | Baseline | +PLT | Baseline | +PLT | Baseline | +PLT | Abs. Δ | Rel. |
| GPT-Micro | 5M | 1.0000 | 1.0000 | 0.1078 | **0.3922** | 0.78 | **0.85** | +0.2844 | ×3.64 |
| GPT-Nano | 2.5M | 1.0000 | 1.0000 | 0.0000 | 0.0000 | 0.70 | 0.70 | 0.0000 | – |
| GPT-Mini | 10M | 1.0000 | 1.0000 | 0.1386 | **0.1683** | 0.82 | **0.83** | +0.0297 | ×1.21 |
| Gopher-44M | 44M | 0.9794 | **1.0000** | 0.0097 | **0.1359** | 0.85 | **0.90** | +0.1262 | ×14.0 |
| OpenAI-GPT | 117M | 0.9697 | **1.0000** | 0.0000 | **0.0099** | 0.88 | **0.93** | +0.0099 | ∞ |
| GPT-2 | 124M | 0.9906 | 0.9906 | 0.0213 | **0.1489** | 0.89 | **0.90** | +0.1276 | ×6.99 |
| GPT-2-Medium | 355M | 0.9794 | **1.0000** | 0.0000 | 0.0000 | 0.90 | **0.93** | 0.0000 | – |
| GPT-2-Large | 774M | 1.0000 | 0.9703 | 0.0000 | **0.0101** | 0.91 | 0.87 | +0.0101 | ∞ |

## D.2   ABLATION ON NEIGHBORHOOD SIZE $k$

We study the sensitivity of our method to the neighborhood size $k$ used for retrieving historical policies / constructing local neighborhood relations in Wasserstein space. Figure 4 reports results on ATARI ALIEN with $k \in \{5, 10, 20, 50\}$. Overall, performance improves as $k$ increases: $k=20$ and $k=50$ consistently outperform smaller neighborhoods, with $k=50$ achieving the best final return. In contrast, very small neighborhoods (e.g., $k=5$) yield weaker late-stage performance, suggesting that overly sparse local graphs provide noisy or incomplete estimates of the underlying policy geometry, which reduces the effectiveness of Laplacian-style regularization and neighborhood-based updates. Notably, compared to the baseline, our method continues to improve beyond mid training, whereas the baseline tends to plateau. These observations support the intuition that reliable local neighborhood structure in Wasserstein space is important for our approach. In practice, we find that moderate-to-large $k$ (e.g., 20) provides a favorable trade-off between performance and computation.

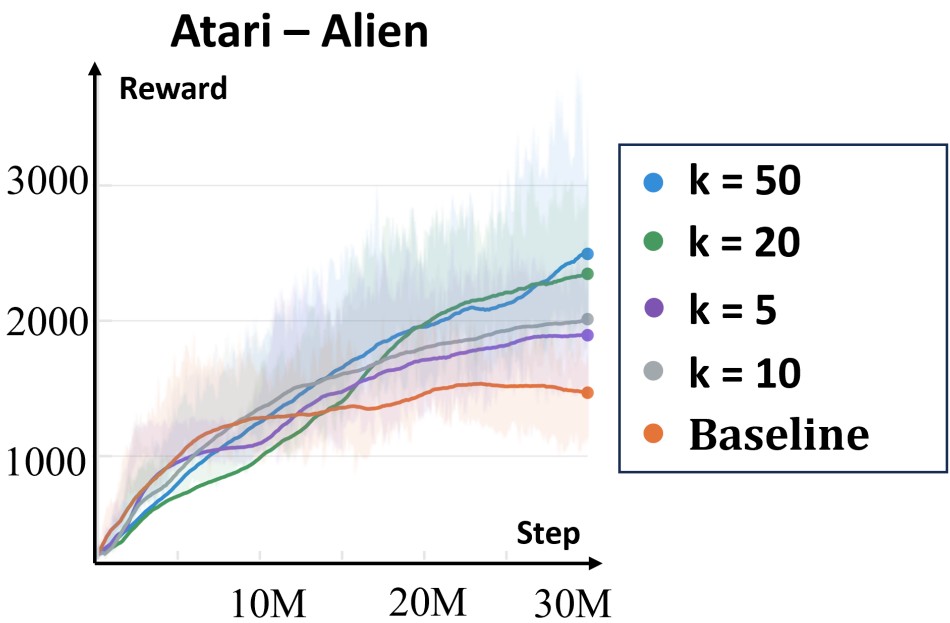

Figure 4: Ablation on the neighborhood size $k$ on ATARI ALIEN. Curves show mean evaluation return over training steps; shaded areas indicate variability across seeds.

### D.3 LOW-DIMENSIONALITY AND MANIFOLD DIAGNOSTICS

**Motivation.** A central premise of this work is that, for each input $x$, the realized policy trajectory $\{\pi_{\theta_t}(\cdot \mid x)\}_t$ does not fill the ambient parameter space but instead concentrates on an *experience-induced low-dimensional subset* of $\mathcal{P}_2(\mathcal{A})$ (idealized as a submanifold). To empirically substantiate this premise and to quantify the geometric effect of PLT, we conduct diagnostics on policy snapshots along training and report three complementary geometry indicators: (i) *effective dimension*, (ii) *local (intrinsic) dimension*, and (iii) *trajectory curvature* in Wasserstein space.[1]

**Evidence for experience-induced low-dimensional structure.** Despite operating in an ambient parameter space of roughly $10^6$ dimensions, the policy trajectory exhibits a markedly low effective dimension: the estimated effective dimension is approximately $46$, corresponding to a compression ratio exceeding $2 \times 10^4$:1. This provides direct empirical support that the learned policies concentrate on a low-dimensional structure, consistent with the "experience-induced Wasserstein policy manifold" hypothesis.

**Geometry regularization induced by PLT.** We compare the above diagnostics between the baseline training and +PLT. Across matched training budgets, PLT consistently yields a *smoother* and *more concentrated* trajectory in Wasserstein space:

- **Effective dimension:** decreases by $0.4\%$ (paired $t$-test: $t(31) = 2.98$, $p = 0.007$), indicating slightly stronger concentration of policy iterates.

- **Trajectory curvature:** decreases by $9.9\%$ ($t(31) = 3.12$, $p = 0.004$), suggesting that PLT substantially suppresses jagged, geometry-unaware jumps and promotes smoother policy evolution.

- **Local (intrinsic) dimension:** decreases by $4.7\%$ ($t(31) = 0.93$, $p = 0.360$); while not statistically significant at $\alpha = 0.05$, the trend is directionally consistent with the above measures.

---

[1]We report mean $\pm$ s.d. over seeds and perform paired two-sided $t$-tests at $\alpha = 0.05$.

**Statistical testing protocol.** All tests use paired, two-sided $t$-tests across $n = 32$ paired runs (degrees of freedom 31), with significance level $\alpha = 0.05$. The resulting statistics are:

- Effective dimension: $t(31) = 2.98$, $p = 0.007$.
- Trajectory curvature: $t(31) = 3.12$, $p = 0.004$.
- Local dimension: $t(31) = 0.93$, $p = 0.360$.

**Takeaway.** Together, these diagnostics provide converging empirical evidence that (i) policy learning trajectories concentrate on a low-dimensional structure in Wasserstein space, and (ii) PLT acts as a Laplacian-style geometric regularizer that makes this structure *more coherent* (lower effective dimension) and the trajectory *smoother* (lower curvature). This supports the paper's key claim that one can *learn and exploit policy geometry from experience* rather than relying solely on parameter- or KL-scale control.

# E  OVERHEAD OF PLT: MEMORY AND COMPUTATION

**Scope.**  This appendix summarizes the additional *memory* and *compute* overhead introduced by Policy Laplacian Trace (PLT) relative to the base optimizer, separating (i) RL/MDP policies and (ii) LLMs-as-policies. We emphasize that PLT is modular: its overhead is controlled by the trace capacity, the stored representation, and the OT approximation used for fusion.

## E.1  RL/MDP SETTING

**Storage.**  With trace capacity $M$, PLT stores (i) a state embedding $z(s) \in \mathbb{R}^{d_z}$ and (ii) a compact representation of the policy at that state (e.g., distribution parameters) of dimension $d_\phi$. The asymptotic storage is

$$\mathcal{O}\big(M(d_z + d_\phi)\big), \qquad d_\phi = \begin{cases} 2d_a, & \text{continuous actions (e.g., Gaussian mean/scale)} \\ |\mathcal{A}|, & \text{discrete actions (probability vector).} \end{cases} \tag{43}$$

**Example.**  For $M{=}10^4$, $d_z{=}64$, $d_a{=}6$ (continuous), the additional footprint is $10^4(64 + 2 \cdot 6) = 7.6 \times 10^5$ floats $\approx 3.0\,\text{MB}$ with float32.

**Compute.**  Per update, PLT adds (a) neighborhood retrieval on embeddings (e.g., $k$NN) and (b) a small OT/Sinkhorn computation for fusion when enabled. The dominant terms are:

$$\text{retrieval: } \mathcal{O}(Md_z) \text{ (naive)} \quad \text{or sublinear with an ANN index;} \qquad \text{OT: } \mathcal{O}(m^2 I), \tag{44}$$

where $m$ is the number of support points/particles used in the OT approximation (typically 8–32) and $I$ is the number of Sinkhorn iterations. OT-related buffers are *temporary* and do not scale with $M$.

## E.2  LLM SETTING

**Storage.**  For LLMs, naively storing full next-token distributions is prohibitive. If PLT stores $N$ trace entries (e.g., $N \approx L \times K$ for sequence length $L$ and $K$ entries per position) and records both a hidden-state key of dimension $d_h$ and a full vocabulary distribution of size $V$, the storage scales as

$$\mathcal{O}\big(N(d_h + V)\big), \tag{45}$$

which can reach multi-GB for typical $V \approx 50\,\text{k}$. In practice, our implementation uses a *compressed* representation (e.g., a semantic embedding proxy) of dimension $d_{\text{sem}} \approx d_h$ instead of storing full distributions, reducing storage to

$$\mathcal{O}\big(N(d_h + d_{\text{sem}})\big). \tag{46}$$

**Example (GPT-2 Small).**  With $L{=}1024$, $K{=}10$ so $N{=}10{,}240$, $d_h{=}768$, $V{=}50{,}257$: storing full distributions would be $\approx 2.1\,\text{GB}$ (float32), whereas a compressed $(d_h{+}d_{\text{sem}}) \approx 1536$ representation is $\approx 63\,\text{MB}$.

**Compute.**  PLT adds (a) retrieval over keys (per token/position) and (b) fusion. Retrieval is

$$\mathcal{O}(Nd_h) \text{ (naive per query)}, \qquad \mathcal{O}(BTNd_h) \text{ over a batch of size } B \text{ and length } T, \tag{47}$$

and can be reduced with sparse gating (apply PLT only on selected tokens) and ANN indexing. Fusion costs depend on the representation: if operating on full distributions it is $\mathcal{O}(V)$ per fused token; with compressed representations it is $\mathcal{O}(d_{\text{sem}})$.

**Practical overhead (order-of-magnitude).**  Across our settings, the trace typically adds $\sim$10–20% memory relative to model parameters for LLMs when using compressed storage, and a moderate runtime overhead from retrieval and small-batch OT (often $\sim$10–30% depending on gating and $m, I$).

**Key implementation knobs.**  To balance performance and overhead, PLT exposes: (i) fixed-capacity FIFO or similarity-prioritized trace, (ii) compressed policy representations (semantic proxies), (iii) sparse retrieval/fusion via gating, and (iv) small-batch entropic OT with $m \in [8, 32]$.

**Deployment note.**  PLT is designed to be optional at inference: one can disable fusion at test time to recover the base model's latency profile, while retaining PLT benefits during training.

# F THEORETICAL PROOF

**Notation** $\mathcal{S}$ denotes the state space and $\mathcal{A}$ the action space equipped with a norm $\|\cdot\|$. $\mathcal{P}_2(\mathcal{A})$ is the set of Borel probability measures on $\mathcal{A}$ with finite second moment. $W_2(\cdot,\cdot)$ is the 2-Wasserstein distance on $\mathcal{P}_2(\mathcal{A})$, and $W_{2,\varepsilon}$ is its entropic-regularized surrogate (e.g., Sinkhorn OT) with regularization strength $\varepsilon > 0$. A stochastic policy is written as $\pi(\cdot \mid s)$, and when parameterized we write $\pi_\theta(\cdot \mid s)$ with parameter $\theta \in \Theta \subset \mathbb{R}^d$. We use $t$ to index optimization iterations / interaction steps, and $i, j$ to index samples stored in a buffer. The visited (or replay-buffer) state set at time $t$ is $\mathcal{S}_t = \{s_i\}_{i=1}^t$, and we abbreviate $\pi_i := \pi(\cdot \mid s_i)$ (or $\pi_{\theta_i}(\cdot \mid s_i)$ when emphasizing parameters). A state representation for similarity is $\varphi : \mathcal{S} \to \mathbb{R}^m$, $\kappa(\cdot)$ is a kernel, $h > 0$ is its bandwidth, and $k$ denotes the neighborhood size for $k$NN graphs. The experience-induced graph is $G_t = (V_t, E_t)$ with node set $V_t = \mathcal{S}_t$, edge set $E_t$, and symmetric nonnegative weights $w_{ij} = w_{ji} \geq 0$; $\mathcal{N}_k(s_i)$ is the $k$-nearest-neighbor set of node $s_i$ in $G_t$. $\Lambda_s \subset \mathcal{P}_2(\mathcal{A})$ denotes the realizable policy set at state $s$. $\mathrm{KL}(\mu\|\nu)$ is the Kullback–Leibler divergence between measures $\mu$ and $\nu$ (assumed finite when used). $\lambda \geq 0$ is the geometric-regularization coefficient, and $\eta > 0$ is a step size. $\ell_t(\mu)$ denotes a task-driven local objective or surrogate loss functional evaluated at a candidate policy distribution $\mu$. For a measurable map $F : \mathcal{A} \to \mathcal{A}$, $F_{\#}\mu$ denotes the pushforward measure. $\mathrm{Id} : \mathcal{A} \to \mathcal{A}$ is the identity map. For $\mu \in \mathcal{P}_2(\mathcal{A})$, $T_\mu \mathcal{P}_2(\mathcal{A})$ denotes the (Otto) tangent space used in Wasserstein first-variation calculus. $L^2(\mu)$ denotes square-integrable vector fields and $\langle u, v \rangle_{L^2(\mu)} := \int u(a)^\top v(a)\, d\mu(a)$. We write $\mathrm{Log}_\mu(\nu)$ for the (possibly set-valued) Wasserstein logarithm direction; when the OT map $T_{\mu \to \nu}$ is unique, $\mathrm{Log}_\mu(\nu) = T_{\mu \to \nu} - \mathrm{Id}$.

## F.1 EXPERIENCE-INDUCED GEOMETRY AND THE WASSERSTEIN POLICY OPERATOR

**Scope and purpose** This subsection fixes the mathematical objects used throughout the paper. The goal is to make precise (i) *what* the policy distributions live on, (ii) *what* geometric structure is induced by experience, and (iii) *how* the proposed update can be viewed as a proximal/geometric step. Proofs are deferred unless explicitly stated.

(see, e.g., Ambrosio et al. (2005); Villani et al. (2008) for the first-variation formula and Otto calculus; Cuturi (2013); Peyré et al. (2019) for entropic OT smoothness and dual potentials).

**Definition F.1** (Policy as a point in Wasserstein space). Let $(\mathcal{A}, \|\cdot\|)$ be the action space and $\mathcal{P}_2(\mathcal{A})$ the set of Borel probability measures with finite second moment. A (stochastic) policy is a state-indexed family of measures

$$\pi(\cdot \mid s) \in \mathcal{P}_2(\mathcal{A}), \qquad s \in \mathcal{S}, \tag{48}$$

equipped with the 2-Wasserstein metric $W_2$ on $\mathcal{P}_2(\mathcal{A})$. Optionally, we use the entropic OT surrogate $W_{2,\varepsilon}$ as a computationally tractable approximation of $W_2$.

**Assumption F.2** (Realizable policy family as a benign finite-dimensional set). Let $\Theta \subset \mathbb{R}^d$ be a parameter space and $\{\pi_\theta(\cdot \mid s)\}_{\theta \in \Theta}$ a parameterized policy family. For each state $s$, define the realizable set

$$\Lambda_s := \big\{\pi_\theta(\cdot \mid s) : \theta \in \Theta\big\} \subset \mathcal{P}_2(\mathcal{A}). \tag{49}$$

We assume $\Lambda_s$ is locally well-behaved (e.g., a finite-dimensional $C^1$ embedded submanifold of $\mathcal{P}_2(\mathcal{A})$ or admits a locally smooth chart/retraction), so that local geodesic approximations and projections/retractions onto $\Lambda_s$ are meaningful.

**Definition F.3** (Experience-induced state graph). Let $\mathcal{S}_t = \{s_i\}_{i=1}^t$ denote the set of visited states (or a replay buffer subset) and let $\varphi : \mathcal{S} \to \mathbb{R}^m$ be a state representation used for similarity. We build an undirected (typically $k$NN) graph $G_t = (V_t, E_t)$ with nodes $V_t = \mathcal{S}_t$ and symmetric weights

$$w_{ij} \geq 0, \qquad w_{ij} = 0 \text{ if } (i, j) \notin E_t, \qquad w_{ij} = w_{ji}, \tag{50}$$

e.g., $w_{ij} = \kappa(\|\varphi(s_i) - \varphi(s_j)\|/h)$ for a kernel $\kappa$ and bandwidth $h$. We write $\mathcal{N}_k(s_i)$ for the neighborhood of $s_i$ in $G_t$.

**Definition F.4** (Experience-induced policy Dirichlet energy). Associate each visited state $s_i$ with its policy measure $\pi_i := \pi(\cdot \mid s_i) \in \mathcal{P}_2(\mathcal{A})$. The (quadratic) memory/geometry energy is the Wasserstein graph Dirichlet energy

$$E_{\mathrm{mem}}(\{\pi_i\}_{i=1}^t) := \frac{1}{2} \sum_{i,j=1}^t w_{ij}\, W_2^2(\pi_i, \pi_j). \tag{51}$$

When using entropic OT, we replace $W_2$ by $W_{2,\varepsilon}$. More generally, the $p$-Dirichlet version is

$$E_{\mathrm{mem}}^{(p)}(\{\pi_i\}) := \frac{1}{p}\sum_{i,j} w_{ij}\, W_2^p(\pi_i, \pi_j), \qquad p \geq 1, \tag{52}$$

which induces a nonlinear (graph) $p$-Laplacian geometry.

**Lemma F.5** (First variation induces a Wasserstein graph Laplacian). *Let $\mathcal{A} \subset \mathbb{R}^d$ be convex and equip $\mathcal{P}_2(\mathcal{A})$ with the quadratic-cost 2-Wasserstein metric $W_2$ induced by $c(a,b) = \|a - b\|_2^2$. Consider the weighted Dirichlet energy on a collection of policies $\{\pi_\ell\}_{\ell=1}^n \subset \mathcal{P}_2(\mathcal{A})$:*

$$E_{\mathrm{mem}}(\{\pi_\ell\}_{\ell=1}^n) := \frac{1}{2}\sum_{\ell,m=1}^n w_{\ell m}\, W_2^2(\pi_\ell, \pi_m), \qquad w_{\ell m} = w_{m\ell} \geq 0. \tag{53}$$

*Fix a node $i$ and assume:*

*(1) $\pi_i$ is absolutely continuous w.r.t. Lebesgue measure on $\mathcal{A}$;*

*(2) for every $j$ with $w_{ij} > 0$, the optimal transport from $\pi_i$ to $\pi_j$ is induced by an (a.e.) unique map $T_{i\to j} : \mathcal{A} \to \mathcal{A}$.*

*Define the (single-valued) Wasserstein logarithm direction*

$$\log_{\pi_i}(\pi_j) := T_{i\to j} - \mathrm{Id} \in L^2(\pi_i). \tag{54}$$

*Let $v \in T_{\pi_i}\mathcal{P}_2(\mathcal{A})$ and consider the perturbation $\pi_i^\tau := (\mathrm{Id} + \tau v)_{\#}\pi_i$ for $\tau$ small.*

*Then the directional derivative of $E_{\mathrm{mem}}$ along $v$ exists and satisfies*

$$\frac{d}{d\tau}\bigg|_{\tau=0} E_{\mathrm{mem}}(\pi_1, \dots, \pi_i^\tau, \dots, \pi_n) = -\bigg\langle \sum_{j=1}^n w_{ij}\, \log_{\pi_i}(\pi_j),\, v \bigg\rangle_{L^2(\pi_i)}. \tag{55}$$

*Equivalently, the Wasserstein–Riemannian (Otto) gradient of $E_{\mathrm{mem}}$ with respect to $\pi_i$ is*

$$\nabla_{\pi_i}^{W_2} E_{\mathrm{mem}} = -\sum_{j=1}^n w_{ij}\, \log_{\pi_i}(\pi_j) = \sum_{j=1}^n w_{ij}\, (\mathrm{Id} - T_{i\to j}). \tag{56}$$

*Proof.* Write $E_{\mathrm{mem}} = \frac{1}{2}\sum_{\ell,m} w_{\ell m} W_2^2(\pi_\ell, \pi_m)$ and differentiate only the terms that depend on $\pi_i$. Using symmetry $w_{\ell m} = w_{m\ell}$ and $W_2(\pi_\ell, \pi_m) = W_2(\pi_m, \pi_\ell)$, we can rewrite the $\pi_i$-dependent part as

$$E_{\mathrm{mem}}(\pi_1, \dots, \pi_i, \dots, \pi_n) = \sum_{j=1}^n w_{ij}\, W_2^2(\pi_i, \pi_j)\ +\ \text{(terms independent of } \pi_i\text{)}. \tag{57}$$

Hence,

$$\frac{d}{d\tau}\bigg|_{\tau=0} E_{\mathrm{mem}}(\pi_1, \dots, \pi_i^\tau, \dots, \pi_n) = \sum_{j=1}^n w_{ij}\, \frac{d}{d\tau}\bigg|_{\tau=0} W_2^2(\pi_i^\tau, \pi_j). \tag{58}$$

We now invoke the standard Otto-calculus first-variation identity for the squared 2-Wasserstein distance (see, e.g., , Ch. 8–10 Villani et al. (2008)): under Assumptions (1)–(2), the functional $\mu \mapsto \frac{1}{2}W_2^2(\mu, \nu)$ has Wasserstein (Otto) gradient at $\mu$ equal to

$$\nabla_\mu^{W_2}\left(\frac{1}{2}W_2^2(\mu, \nu)\right) = -\log_\mu(\nu) = \mathrm{Id} - T_{\mu\to\nu}, \tag{59}$$

and consequently, for the perturbation $\mu^\tau = (\mathrm{Id} + \tau v)_{\#}\mu$,

$$\frac{d}{d\tau}\bigg|_{\tau=0}\left(\frac{1}{2}W_2^2(\mu^\tau, \nu)\right) = \left\langle \nabla_\mu^{W_2}\left(\tfrac{1}{2}W_2^2(\mu, \nu)\right),\, v \right\rangle_{L^2(\mu)} = -\langle \log_\mu(\nu), v\rangle_{L^2(\mu)}. \tag{60}$$

Applying equation 60 with $\mu = \pi_i$ and $\nu = \pi_j$ gives

$$\frac{d}{d\tau}\bigg|_{\tau=0} W_2^2(\pi_i^\tau, \pi_j) = -2\langle \log_{\pi_i}(\pi_j),\, v\rangle_{L^2(\pi_i)}. \tag{61}$$

Substituting equation 61 into equation 58 yields

$$\frac{d}{d\tau}\bigg|_{\tau=0} E_{\mathrm{mem}}(\pi_1,\ldots,\pi_i^\tau,\ldots,\pi_n) = -2\Big\langle \sum_{j=1}^n w_{ij} \log_{\pi_i}(\pi_j),\, v\Big\rangle_{L^2(\pi_i)}. \tag{62}$$

Finally, note that our definition of $E_{\mathrm{mem}}$ contains a prefactor $\frac{1}{2}$, while the $\pi_i$-dependent part in equation 57 contributes two symmetric copies (the $(i, j)$ and $(j, i)$ terms). These cancel the factor 2 above, yielding exactly

$$\frac{d}{d\tau}\bigg|_{\tau=0} E_{\mathrm{mem}}(\pi_1,\ldots,\pi_i^\tau,\ldots,\pi_n) = -\Big\langle \sum_{j=1}^n w_{ij} \log_{\pi_i}(\pi_j),\, v\Big\rangle_{L^2(\pi_i)}. \tag{63}$$

By the Riesz representation of the differential in the Otto metric, this identifies the Wasserstein gradient as $\nabla^{W_2}_{\pi_i} E_{\mathrm{mem}} = -\sum_j w_{ij} \log_{\pi_i}(\pi_j) = \sum_j w_{ij}(\mathrm{Id} - T_{i\to j})$.     $\square$

*Remark* F.6 (Non-uniqueness and discrete policies: subgradients / entropic smoothing). If the OT map from $\pi_i$ to $\pi_j$ is not unique, the Wasserstein logarithmic direction becomes set-valued (equivalently, the optimal coupling may be non-unique), so the Otto gradient in Lemma F.5 should be interpreted in a subdifferential sense (Ambrosio et al., 2005; Divol et al., 2025). Let $\partial^{W_2}_{\pi_i}$ denote the metric (Wasserstein) subdifferential. Then one can replace Lemma F.5 by the inclusion

$$\partial^{W_2}_{\pi_i} E_{\mathrm{mem}} \ni -\sum_{j=1}^n w_{ij}\, \xi_{i\to j}, \qquad \xi_{i\to j} \in \log_{\pi_i}(\pi_j), \tag{64}$$

i.e., any selection of optimal couplings (or optimal maps when they exist) yields a valid element of the subdifferential and hence a valid descent direction for the Wasserstein gradient flow.

For discrete policies (finite action space), a convenient alternative is to replace $W_2^2$ by the entropic OT surrogate $W_{2,\varepsilon}^2$ with $\varepsilon > 0$. In a discrete action space, $W_{2,\varepsilon}^2$ is everywhere differentiable with respect to the probability vector on the simplex, and its gradient is given by the (centered) Sinkhorn dual potentials returned by the Sinkhorn algorithm (equivalently, by the gradient of the entropic OT dual objective with respect to the source marginal) (Cuturi, 2013; Peyré et al., 2019). Consequently, the entropic energy

$$E_{\mathrm{mem}}^\varepsilon(\{\pi_\ell\}) := \frac{1}{2} \sum_{\ell,m=1}^n w_{\ell m}\, W_{2,\varepsilon}^2(\pi_\ell, \pi_m) \tag{65}$$

is differentiable on the simplex, and the resulting "Laplacian force" at node $i$ is simply the sum of the neighbor-wise entropic OT gradients:

$$\nabla_{\pi_i} E_{\mathrm{mem}}^\varepsilon = \sum_{j=1}^n w_{ij}\, \nabla_{\pi_i}\left(\frac{1}{2}\, W_{2,\varepsilon}^2(\pi_i, \pi_j)\right), \tag{66}$$

which directly matches the gradients implicitly computed by our Sinkhorn-based OT-fuse implementation.

**Definition F.7** (Policy Laplacian Trace (PLT) as an update map). Fix a time $t$ and a query state $s_t$ with current policy $\pi_t(\cdot \mid s_t)$. Let $\mathcal{N}_k(s_t)$ be its neighborhood in the experience-induced graph, with corresponding neighbor policies $\{\pi_j(\cdot \mid s_j)\}_{j\in\mathcal{N}_k(s_t)}$ and weights $\{w_{tj}\}$. Given a task-driven local signal denoted by $\ell_t(\cdot)$, define the fused/updated policy distribution as

$$\pi_t^{\mathrm{fuse}} \in \arg\min_{\mu\in\Lambda_{s_t}}\left\{\ell_t(\mu) + \frac{\lambda}{2} \sum_{j\in\mathcal{N}_k(s_t)} w_{tj}\, W_2^2(\mu, \pi_j) + \frac{1}{\eta}\,\mathrm{KL}(\mu\|\pi_t)\right\}. \tag{67}$$

We define the operator

$$\mathrm{PLT}_t:\ (\pi_t, \{\pi_j\}_{j\in\mathcal{N}_k(s_t)}, \ell_t) \mapsto \pi_t^{\mathrm{fuse}}. \tag{68}$$

**Lemma F.8** (PLT is the implicit Euler (proximal) step of a composite geometric energy). *Fix an iteration $t$ and a state $s_t$. Let $\Lambda_{s_t}$ be a nonempty, closed, convex subset of the probability simplex $\Delta^{m-1}$ (finite action space with $m$ actions).[2] Let $\pi_t \in \Lambda_{s_t}$ be the current policy at $s_t$.*

*Let $\ell_t : \Lambda_{s_t} \to \mathbb{R} \cup \{+\infty\}$ be proper, lower semicontinuous, and convex. Let $\{(\pi_i, w_i)\}_{i \in \mathcal{N}_t}$ be the retrieved neighbour policies with weights $w_i \geq 0$ and $\sum_{i \in \mathcal{N}_t} w_i = 1$. Let $d : \Lambda_{s_t} \times \Lambda_{s_t} \to \mathbb{R}_+$ be a metric such that the map $\pi \mapsto \frac{1}{2}d^2(\pi, \pi_i)$ is convex on $\Lambda_{s_t}$ for each $i \in \mathcal{N}_t$ (e.g., $d = W_{2,\varepsilon}$ with fixed ground cost and $\varepsilon > 0$). Define the memory potential*

$$\Phi_t(\pi) \; := \; \frac{1}{2} \sum_{i \in \mathcal{N}_t} w_i \, d^2(\pi, \pi_i), \qquad \mathcal{J}_t(\pi) \; := \; \ell_t(\pi) + \lambda \, \Phi_t(\pi),$$

*for some $\lambda \geq 0$.*

*For any stepsize $\eta > 0$, define the PLT update as the unique minimizer*

$$\pi_{t+1} \; := \; \arg \min_{\pi \in \Lambda_{s_t}} \left\{ \mathcal{J}_t(\pi) \; + \; \frac{1}{\eta} D_{\mathrm{KL}}(\pi \| \pi_t) \right\}. \tag{69}$$

*Then:*

1. *(**Existence and uniqueness**) The minimization problem equation 69 admits a unique solution $\pi_{t+1} \in \Lambda_{s_t}$.*

2. *(**Implicit Euler / proximal step**) $\pi_{t+1}$ is exactly the implicit Euler step of the $D_{\mathrm{KL}}$-gradient flow of $\mathcal{J}_t$ on $\Lambda_{s_t}$, i.e., it satisfies the first-order optimality condition*

$$0 \in \partial \mathcal{J}_t(\pi_{t+1}) \; + \; \frac{1}{\eta} \, \partial_\pi D_{\mathrm{KL}}(\pi \| \pi_t)\big|_{\pi = \pi_{t+1}} \; + \; N_{\Lambda_{s_t}}(\pi_{t+1}), \tag{70}$$

   *where $N_{\Lambda_{s_t}}(\cdot)$ denotes the normal cone of $\Lambda_{s_t}$.*

3. *(**Trust region in distribution space**) Moreover,*

$$\mathcal{J}_t(\pi_{t+1}) + \frac{1}{\eta} D_{\mathrm{KL}}(\pi_{t+1} \| \pi_t) \; \leq \; \mathcal{J}_t(\pi_t), \tag{71}$$

   *so the step enforces a $D_{\mathrm{KL}}$-proximal trust region while decreasing the composite energy.*

**Corollary F.9** (Small-step expansion: task drift + Laplacian-type smoothing). *Assume in addition that $\Lambda_{s_t}$ is a smooth Riemannian manifold endowed with a distance $d$ such that, on a geodesically convex neighbourhood $\mathcal{U} \subset \Lambda_{s_t}$ containing $\pi_t$, each $\pi_i \in \mathcal{U}$ is connected to $\pi$ by a unique minimizing geodesic and*

$$\nabla_\pi \left( \frac{1}{2} d^2(\pi, \pi_i) \right) \; = \; -\log_\pi(\pi_i), \qquad \forall \pi \in \mathcal{U}, \tag{72}$$

*where $\log_\pi(\pi_i) \in T_\pi \Lambda_{s_t}$ denotes the Riemannian logarithm map. Assume $\ell_t$ is $C^1$ on $\mathcal{U}$ and $\pi_{t+1} \in \mathcal{U}$.*

*Then $\pi_{t+1}$ satisfies the implicit equation*

$$\frac{1}{\eta} \operatorname{grad}_{D_{\mathrm{KL}}} D_{\mathrm{KL}}(\cdot \| \pi_t)\big|_{\pi_{t+1}} \; + \; \operatorname{grad}_{D_{\mathrm{KL}}} \ell_t(\pi_{t+1}) \; - \; \lambda \sum_{i \in \mathcal{N}_t} w_i \log_{\pi_{t+1}}(\pi_i) \; = \; 0, \tag{73}$$

*and, as $\eta \to 0$, admits the first-order expansion in $T_{\pi_t} \Lambda_{s_t}$:*

$$\pi_{t+1} = \pi_t - \eta \left( \operatorname{grad}_{D_{\mathrm{KL}}} \ell_t(\pi_t) \; - \; \lambda \sum_{i \in \mathcal{N}_t} w_i \log_{\pi_t}(\pi_i) \right) + o(\eta). \tag{74}$$

*In particular, the memory term $\sum_i w_i \log_{\pi_t}(\pi_i)$ is a Laplacian-type smoothing direction: it points from $\pi_t$ towards the (weighted) barycentric neighbourhood in the geometry induced by $d$.*

---

[2] For Gaussian policies, replace $\Delta^{m-1}$ by the parameter manifold and interpret $D_{\mathrm{KL}}$ as a Bregman divergence induced by the log-partition; the statement remains identical.

**Remark (operator vs. storage).** Definitions F.4–F.7 emphasize that memory enters learning through an *operator* (a variational update rule) rather than a particular storage mechanism: any buffer that provides $\{(s_i, \pi(\cdot \mid s_i))\}$ together with a graph construction yields the same mathematical object.

## F.2   METRIC MISMATCH: KL DIVERGENCE AND WASSERSTEIN DISTANCE

To motivate the choice of the Wasserstein metric for our analysis, we first demonstrate a fundamental limitation of the standard KL divergence trust region. Specifically, we show that constraining the KL divergence does not imply a bound on the Wasserstein-2 displacement, which is the geometric quantity governing the transport cost.

**Lemma F.10** (KL does not control $W_2$ without additional structure)**.** *There is no universal constant $C > 0$ such that $W_2(\nu, \mu) \leq C\sqrt{D_{\mathrm{KL}}(\nu\|\mu)}$ holds for all pairs $(\nu, \mu) \in \mathcal{P}_2(\mathbb{R}^d)$. In particular, one can construct a sequence of (Gaussian) measures with $D_{\mathrm{KL}}(\nu_n\|\mu_n) \to 0$ while $W_2(\nu_n, \mu_n) \to \infty$.*

**Remark (When KL *does* control $W_2$).** Such a control can hold under additional regularity/compactness assumptions, e.g., when the reference measure satisfies a $T_2$/log-Sobolev-type condition, or when supports / second moments are uniformly bounded, yielding bounds of the form $W_2^2(\nu, \mu) \lesssim D_{\mathrm{KL}}(\nu\|\mu)$ with problem-dependent constants. Our lemma rules out only a *structure-free* (universal) control.

*Proof.* We prove this by constructing an explicit counterexample using Gaussian distributions with shifting means and expanding variances. For each $n \in \mathbb{N}$, define:

$$P_n = \mathcal{N}(n, n^4), \qquad Q_n = \mathcal{N}(0, n^4). \tag{75}$$

Note that both distributions share the same variance $\sigma_n^2 = n^4$ but have means separated by $\mu_{P_n} - \mu_{Q_n} = n$.

**1.   KL divergence vanishes**   The KL divergence between two univariate Gaussians $P = \mathcal{N}(\mu_1, \sigma^2)$ and $Q = \mathcal{N}(\mu_2, \sigma^2)$ is given by the formula $D_{\mathrm{KL}}(P\|Q) = \frac{(\mu_1-\mu_2)^2}{2\sigma^2}$. Substituting the parameters for our sequence:

$$D_{\mathrm{KL}}(P_n\|Q_n) = \frac{(n-0)^2}{2 \cdot n^4} = \frac{n^2}{2n^4} = \frac{1}{2n^2}. \tag{76}$$

Clearly, as $n \to \infty$,

$$D_{\mathrm{KL}}(P_n\|Q_n) \to 0. \tag{77}$$

**2. Wasserstein distance diverges.**   The squared Wasserstein-2 distance between two Gaussians is given by $W_2^2(\mathcal{N}(\mu_1, \sigma_1^2), \mathcal{N}(\mu_2, \sigma_2^2)) = (\mu_1-\mu_2)^2 + (\sigma_1-\sigma_2)^2$. Since $\sigma_1 = \sigma_2 = n^2$, the variance term vanishes, and the distance is purely determined by the shift in means:

$$W_2^2(P_n, Q_n) = (n-0)^2 + (n^2 - n^2)^2 = n^2. \tag{78}$$

Taking the square root, we have:

$$W_2(P_n, Q_n) = n. \tag{79}$$

Thus, as $n \to \infty$,

$$W_2(P_n, Q_n) \to +\infty. \tag{80}$$

**Conclusion.**   Suppose, for the sake of contradiction, that a constant $C$ exists such that $W_2(P, Q) \leq C\sqrt{D_{\mathrm{KL}}(P\|Q)}$ for all pairs. Applying this to our sequence would imply:

$$n \leq C\sqrt{\frac{1}{2n^2}} = \frac{C}{\sqrt{2}n} \implies n^2 \leq \frac{C}{\sqrt{2}}. \tag{81}$$

This inequality clearly fails for sufficiently large $n$. The construction demonstrates that when the reference distribution $Q_n$ becomes sufficiently "flat" (large variance), the KL divergence becomes insensitive to significant shifts in the mean, whereas the Wasserstein distance correctly captures the geometric displacement. □

### F.3  GEOMETRIC OPTIMIZATION PRINCIPLE OF OT-FUSE

We now provide a theoretical foundation for the OT-fuse module. While the fusion rule (constructing the mixture coupling) is computationally efficient, we show here that it formally constitutes an approximate proximal update step on the Wasserstein manifold. This characterizes OT-fuse not as a heuristic, but as a principled geometric optimization procedure.

**Proposition F.11** (OT-Fuse as an explicit Euler step for the entropic Wasserstein gradient flow)**.** *Let* $\mathcal{A} = \{1, \ldots, m\}$ *be a finite action set and let* $\Delta^{m-1}$ *denote the probability simplex. Fix* $\varepsilon > 0$ *and equip* $\Delta^{m-1}$ *with the entropic OT transport cost* $W_{2,\varepsilon}^2(\cdot, \cdot)$ *induced by a ground cost matrix* $C \in \mathbb{R}_+^{m \times m}$. *Let* $\{\pi_i\}_{i \in \mathcal{N}}$ *be retrieved neighbour policies with weights* $w_i \geq 0$, $\sum_{i \in \mathcal{N}} w_i = 1$. *Define the memory potential*

$$\Phi_t(\pi) := \frac{1}{2} \sum_{i \in \mathcal{N}} w_i \, W_{2,\varepsilon}^2(\pi, \pi_i), \qquad \pi \in \Delta^{m-1}. \tag{82}$$

**(i) Explicit Euler interpretation.**  *Assume* $\pi_t \in \mathrm{int}(\Delta^{m-1})$. *Then* $\Phi_t$ *is continuously differentiable on* $\mathrm{int}(\Delta^{m-1})$ *and the gradient flow*

$$\dot{\pi}(\tau) = -\nabla\Phi_t(\pi(\tau)) \tag{83}$$

*is well-defined locally. Moreover, the OT-Fuse update*

$$\pi_t^{\mathrm{fuse}} := \pi_t - \tau \sum_{i \in \mathcal{N}} w_i \, \nabla_\pi \left( \frac{1}{2} W_{2,\varepsilon}^2(\pi, \pi_i) \right) \Big|_{\pi = \pi_t} \tag{84}$$

*implements a single* explicit Euler step *of equation 83 with stepsize* $\tau > 0$.

**(ii) First-order approximation to the proximal point (implicit Euler).**  *Define the entropic Wasserstein proximal point of* $\Phi_t$ *at* $\pi_t$ *by*

$$\pi^\star := \arg \min_{\pi \in \Delta^{m-1}} \left\{ \frac{1}{2\tau} W_{2,\varepsilon}^2(\pi, \pi_t) + \Phi_t(\pi) \right\}. \tag{85}$$

*Assume in addition that* $\nabla\Phi_t$ *is locally Lipschitz in a neighbourhood of* $\pi_t$ *(with constant $L$) and that the minimizer* $\pi^\star$ *exists in* $\mathrm{int}(\Delta^{m-1})$. *Then, for sufficiently small* $\tau$,

$$\|\pi_t^{\mathrm{fuse}} - \pi^\star\| \leq c \tau^2, \tag{86}$$

*for some constant $c$ depending on $L$ and local smoothness constants of* $W_{2,\varepsilon}^2$. *Consequently,* $\pi_t^{\mathrm{fuse}}$ *is a first-order approximation to the implicit proximal point* $\pi^\star$.

**(iii) Link to Sinkhorn dual potentials (algorithmic correspondence).**  *For each neighbour* $\pi_i$, *let* $(f_{t \to i}^\varepsilon, g_{t \to i}^\varepsilon)$ *be the optimal entropic OT dual potentials between* $(\pi_t, \pi_i)$. *Then on* $\mathrm{int}(\Delta^{m-1})$,

$$\nabla_\pi \left( \frac{1}{2} W_{2,\varepsilon}^2(\pi, \pi_i) \right) \Big|_{\pi = \pi_t} = \mathrm{center}(f_{t \to i}^\varepsilon), \tag{87}$$

*where* $\mathrm{center}(\cdot)$ *subtracts an additive constant to make the gradient lie in the tangent space of the simplex (sum of coordinates equal to zero). Thus, OT-Fuse can be implemented by computing Sinkhorn dual potentials for each pair* $(\pi_t, \pi_i)$ *and taking their weighted sum, which matches the update rule equation 84.*

*Proof.* **Step 1: Differentiability and the explicit Euler form**     For $\varepsilon > 0$ on a finite action space, the entropic OT cost $W_{2,\varepsilon}^2(\cdot, \nu)$ is $C^1$ on $\mathrm{int}(\Delta^{m-1})$ (and in fact smooth) with respect to the source marginal; moreover its gradient is given by the optimal Sinkhorn dual potential, up to an additive constant (Peyré et al., 2019; Cuturi, 2013, see e.g. Ch. 4). Since $\Phi_t$ is a finite weighted sum of such terms, it is $C^1$ on $\mathrm{int}(\Delta^{m-1})$, and the gradient flow equation 83 is locally well-posed. By definition of $\Phi_t$, we have

$$\nabla\Phi_t(\pi) = \sum_{i \in \mathcal{N}} w_i \, \nabla_\pi \left( \frac{1}{2} W_{2,\varepsilon}^2(\pi, \pi_i) \right), \tag{88}$$

so the explicit Euler discretization of equation 83 with step $\tau$ is exactly equation 84.

**Step 2: Implicit Euler equals the proximal point**     The proximal point equation 85 is the standard implicit Euler discretization of the gradient flow associated with $\Phi_t$ under the metric induced by $W_{2,\varepsilon}$ (i.e., the proximal point algorithm / minimizing movement scheme). Its first-order optimality condition is

$$\frac{1}{\tau}\nabla_\pi\left(\frac{1}{2}W_{2,\varepsilon}^2(\pi,\pi_t)\right)\Big|_{\pi=\pi^\star} + \nabla\Phi_t(\pi^\star) = 0. \tag{89}$$

**Step 3: Explicit vs. implicit: a second-order local discrepancy**     Assume $\nabla\Phi_t$ is $L$-Lipschitz near $\pi_t$ and $\pi^\star$ stays in this neighbourhood. Using a Taylor expansion around $\pi_t$,

$$\nabla\Phi_t(\pi^\star) = \nabla\Phi_t(\pi_t) + O(\|\pi^\star - \pi_t\|). \tag{90}$$

Similarly, smoothness of $W_{2,\varepsilon}^2$ implies

$$\nabla_\pi\left(\tfrac{1}{2}W_{2,\varepsilon}^2(\pi,\pi_t)\right)\Big|_{\pi=\pi^\star} = (\pi^\star - \pi_t) + O(\|\pi^\star - \pi_t\|^2) \tag{91}$$

in local coordinates on the simplex.[3] Substituting these expansions into equation 89 yields

$$\frac{1}{\tau}(\pi^\star - \pi_t) + \nabla\Phi_t(\pi_t) = O(\tau), \tag{92}$$

so $\pi^\star = \pi_t - \tau\nabla\Phi_t(\pi_t) + O(\tau^2)$. Comparing with the explicit Euler step $\pi_t^{\text{fuse}} = \pi_t - \tau\nabla\Phi_t(\pi_t)$ gives equation 86.

**Step 4: Sinkhorn dual potentials implement the gradients**     Finally, equation 87 is the standard sensitivity result for entropic OT: the derivative of the entropic OT objective with respect to the source marginal equals the optimal dual potential (up to a constant), hence after centering it yields a valid gradient in the simplex tangent space (Peyré et al., 2019; Cuturi, 2013). This establishes the algorithmic correspondence. □

### F.4 Theoretical Stability Guarantee: Non-Expansiveness

A central concern in policy fusion is stability: can the fusion process amplify input noise or lead to mode collapse? We answer this negatively by proving that the memory potential is geodesically convex, rendering the OT-fuse operator a contraction mapping in the Wasserstein metric.

**Proposition F.12** (Geodesic Convexity and Contraction). *Let $\mathcal{P}_2(\mathcal{A})$ be the space of probability measures on $\mathbb{R}^d$ with finite second moments. Consider the memory potential $\Phi_t(\pi) = \frac{1}{2}\sum_i w_i W_2^2(\pi,\pi_i)$ with $\sum w_i = 1$.*

*(i) **1-Convexity**: The potential $\Phi_t$ is 1-geodesically strongly convex on $\mathcal{P}_2(\mathcal{A})$.*

*(ii) **Non-Expansiveness**: The proximal operator $\pi \mapsto \operatorname{prox}_{\tau\Phi_t}^{W_2}(\pi)$ is a contraction on $(\mathcal{P}_2(\mathcal{A}), W_2)$ with Lipschitz constant $(1+\tau)^{-1}$. Specifically, for any two policies $\pi, \pi' \in \mathcal{P}_2(\mathcal{A})$:*

$$W_2\left(\operatorname{prox}_{\tau\Phi_t}^{W_2}(\pi),\ \operatorname{prox}_{\tau\Phi_t}^{W_2}(\pi')\right) \le \frac{1}{1+\tau}W_2(\pi,\pi'). \tag{93}$$

*Proof.* **Part (i): Strong Geodesic Convexity**     The squared Wasserstein distance $\nu \mapsto W_2^2(\nu,\mu)$ is known to be 1-geodesically convex (or displacement convex) on $\mathcal{P}_2(\mathbb{R}^d)$ for any fixed $\mu$. Since $\Phi_t$ is a convex combination of such functions ($\sum w_i = 1$), it preserves the convexity modulus. Thus, for any constant-speed geodesic $\gamma : [0,1] \to \mathcal{P}_2(\mathcal{A})$, the following inequality holds strictly unless $\gamma$ is a constant curve:

$$\Phi_t(\gamma(s)) \le (1-s)\Phi_t(\gamma(0)) + s\Phi_t(\gamma(1)) - \frac{1}{2}s(1-s)W_2^2(\gamma(0),\gamma(1)). \tag{94}$$

---

[3]On the simplex, this should be interpreted in the tangent space (sum-to-zero constraint); the centering operation removes the additive constant ambiguity of dual potentials.

**Part (ii): Contraction of the Proximal Map**    Let $T(\pi) = \text{prox}_{\tau\Phi_t}^{W_2}(\pi)$. By definition, $u = T(\pi)$ minimizes the objective functional:

$$J_\pi(\nu) := \frac{1}{2\tau} W_2^2(\nu, \pi) + \Phi_t(\nu). \tag{95}$$

Since $\frac{1}{2\tau} W_2^2(\cdot, \pi)$ is $\frac{1}{\tau}$-strongly convex and $\Phi_t$ is 1-strongly convex, the total objective $J_\pi$ is $\sigma$-strongly convex with modulus $\sigma = 1 + \frac{1}{\tau}$.

Consider two inputs $\pi, \pi'$ and their mapped outputs $u = T(\pi), u' = T(\pi')$. The first-order optimality condition for geodesically convex functionals (characterized by the vanishing Wasserstein subgradient) implies:

$$0 \in \partial J_\pi(u) \quad \text{and} \quad 0 \in \partial J_{\pi'}(u'). \tag{96}$$

Using the strong monotonicity of the subgradients of $\sigma$-strongly convex functions on Wasserstein space, we have:

$$\langle \nabla J_\pi(u) - \nabla J_\pi(u'), \text{Log}_u(u') \rangle_\pi \geq \sigma W_2^2(u, u'), \tag{97}$$

where the inner product is on the tangent bundle. Since $\nabla J_\pi(u) = 0$, this simplifies to:

$$\langle -\nabla J_\pi(u'), \text{Log}_u(u') \rangle \geq \sigma W_2^2(u, u'). \tag{98}$$

Note that $\nabla J_\pi(\nu) = \frac{1}{\tau} \nabla(\frac{1}{2} W_2^2(\nu, \pi)) + \nabla \Phi_t(\nu)$. Substituting this into the inequality:

$$\langle -\frac{1}{\tau} \text{Log}_{u'}(\pi) - \nabla \Phi_t(u'), \text{Log}_u(u') \rangle \geq \sigma W_2^2(u, u'). \tag{99}$$

From the optimality of $u'$ for $J_{\pi'}$, we know $\nabla \Phi_t(u') = -\frac{1}{\tau} \text{Log}_{u'}(\pi')$. Substituting this back:

$$\frac{1}{\tau} \langle \text{Log}_{u'}(\pi') - \text{Log}_{u'}(\pi), \text{Log}_u(u') \rangle \geq \sigma W_2^2(u, u'). \tag{100}$$

By the generalized Cauchy-Schwarz inequality on the tangent space:

$$\frac{1}{\tau} W_2(\pi, \pi') W_2(u, u') \geq \frac{1}{\tau} \langle \text{Log}_{u'}(\pi') - \text{Log}_{u'}(\pi), \text{Log}_u(u') \rangle. \tag{101}$$

Combining the inequalities:

$$\frac{1}{\tau} W_2(\pi, \pi') W_2(u, u') \geq \left(1 + \frac{1}{\tau}\right) W_2^2(u, u'). \tag{102}$$

Dividing by $W_2(u, u')$ (assuming $u \neq u'$) and multiplying by $\tau$:

$$W_2(\pi, \pi') \geq (1 + \tau) W_2(u, u') \implies W_2(u, u') \leq \frac{1}{1 + \tau} W_2(\pi, \pi'). \tag{103}$$

$$\square$$

**Corollary F.13** (Robustness). *Since $\tau > 0$ (and $\tau = 1$ in our implementation), the OT-fuse operator strictly contracts the Wasserstein distance between any perturbed policy and the ideal policy. This guarantees that (1) the fusion process is stable against input noise, and (2) recursive application of the module leads to exponential convergence towards the memory consensus, preventing distributional divergence.*

### F.5   DISCRETE DIRICHLET ENERGY AND LAPLACIAN SMOOTHING

To understand the global effect of the OT-fuse mechanism, we analyze it through the lens of functional analysis on graphs. We define a *discrete Dirichlet energy* over the graph of policies, which measures the total geometric smoothness of the policy field.

**Proposition F.14** (Discrete Dirichlet Form Induces Wasserstein Graph Laplacian). *Let $G = (V, E)$ be an undirected graph with vertices $V = \{1, \ldots, n\}$ and symmetric edge weights $w_{ij} \geq 0$. Let $\{\pi_i\}_{i=1}^n \subset \mathcal{P}_2(\mathcal{A})$ be a collection of policy distributions over a convex action space $\mathcal{A} \subseteq \mathbb{R}^d$. Define the discrete Dirichlet energy on the graph of distributions:*

$$E_{\text{mem}}(\pi_1, \ldots, \pi_n) := \frac{1}{4} \sum_{i,j} w_{ij} W_2^2(\pi_i, \pi_j). \tag{104}$$

*Under the regularity assumptions of Lemma F.5 (absolute continuity and unique OT maps), the first variation of $E_{\mathrm{mem}}$ with respect to $\pi_i$ satisfies:*

$$\frac{\delta E_{\mathrm{mem}}}{\delta \pi_i} = -\sum_j w_{ij}\,\varphi_{ij} + C_i, \tag{105}$$

*where $\varphi_{ij}$ is the Kantorovich potential from $\pi_i$ to $\pi_j$ (i.e., $\nabla \varphi_{ij}$ is the optimal transport map $T_{ij}$), and $C_i$ is a constant. Consequently, the Wasserstein gradient flow of $E_{\mathrm{mem}}$ is:*

$$\partial_t \pi_i = \mathrm{div}\left(\pi_i \sum_j w_{ij}\,\mathrm{Log}_{\pi_i}(\pi_j)\right), \qquad i = 1,\ldots,n, \tag{106}$$

*where $\mathrm{Log}_{\pi_i}(\pi_j) = T_{ij} - \mathrm{id}$.*

*Moreover, the stationary points of (7) are exactly the discrete* Wasserstein harmonic maps *satisfying the graph Laplacian condition:*

$$\sum_j w_{ij}\,\mathrm{Log}_{\pi_i}(\pi_j) = 0 \quad \text{for all } i. \tag{107}$$

*Equivalently, each $\pi_i$ is a* Wasserstein barycenter *of its neighbors with weights $\{w_{ij}\}_{j=1}^n$.*

*Proof.* We compute the first variation of $E_{\mathrm{mem}}$ with respect to a single $\pi_i$. Since $E_{\mathrm{mem}}$ is symmetric and $w_{ij} = w_{ji}$, we have:

$$\frac{\delta E_{\mathrm{mem}}}{\delta \pi_i} = \frac{1}{2}\sum_j w_{ij}\,\frac{\delta}{\delta \pi_i} W_2^2(\pi_i, \pi_j) + \frac{1}{2}\sum_j w_{ji}\,\frac{\delta}{\delta \pi_i} W_2^2(\pi_j, \pi_i). \tag{108}$$

By the symmetry of the Wasserstein distance, $W_2^2(\pi_j, \pi_i) = W_2^2(\pi_i, \pi_j)$, and the first variation of $W_2^2(\cdot, \pi_j)$ at $\pi_i$ is well-known:

$$\frac{\delta}{\delta \pi_i}\left(\frac{1}{2} W_2^2(\pi_i, \pi_j)\right) = -\varphi_{ij}, \tag{109}$$

where $\varphi_{ij}$ is a Kantorovich potential for the transport from $\pi_i$ to $\pi_j$, determined up to an additive constant. Similarly,

$$\frac{\delta}{\delta \pi_i}\left(\frac{1}{2} W_2^2(\pi_j, \pi_i)\right) = -\psi_{ji}, \tag{110}$$

where $\psi_{ji}$ is a potential for the transport from $\pi_j$ to $\pi_i$. However, by duality, $\psi_{ji} = \varphi_{ij}^c$, the c-transform of $\varphi_{ij}$. Under the assumption of unique OT maps, we have $\varphi_{ij}^c = \varphi_{ij}$ up to a constant. Thus, both terms contribute equally, yielding:

$$\frac{\delta E_{\mathrm{mem}}}{\delta \pi_i} = -\sum_j w_{ij}\varphi_{ij} + C_i, \tag{111}$$

where $C_i$ is a constant ensuring the normalization of the variation.

The Wasserstein gradient flow (in the Otto calculus) is then given by the continuity equation:

$$\partial_t \pi_i = \mathrm{div}\left(\pi_i \nabla \frac{\delta E_{\mathrm{mem}}}{\delta \pi_i}\right). \tag{112}$$

Substituting the expression for the first variation, and noting that $\nabla \varphi_{ij} = T_{ij}$ and $\mathrm{Log}_{\pi_i}(\pi_j) = T_{ij} - \mathrm{id}$, we obtain (7).

To see the barycenter interpretation, observe that the stationary condition $\partial_t \pi_i = 0$ implies $\sum_j w_{ij}\mathrm{Log}_{\pi_i}(\pi_j) = 0$ (in the tangent space at $\pi_i$). This is precisely the first-order condition for $\pi_i$ being a barycenter of $\{\pi_j\}$ with weights $\{w_{ij}\}$ in the Wasserstein space Cuturi (2013). $\qquad\square$

*Remark* F.15 (Laplacian smoothing of policy fields). Equation (7) can be viewed as a *nonlinear heat equation* on the graph of distributions. Indeed, linearizing around a uniform policy $\bar{\pi}$ (i.e., setting $\pi_i = \bar{\pi} + \varepsilon \rho_i$ with small $\varepsilon$), and using the Taylor expansion $\mathrm{Log}_{\pi_i}(\pi_j) \approx \nabla(\psi_j - \psi_i)$ for some potentials $\psi_i$, we recover the classical graph Laplacian acting on the potentials:

$$\partial_t \psi_i \approx \sum_j w_{ij}(\psi_j - \psi_i). \tag{113}$$

Thus, $E_{\mathrm{mem}}$ induces a smoothing flow that dissipates variations between neighboring policies, analogous to Laplacian smoothing of scalar functions on graphs.

**Corollary F.16** (Stabilization via Dirichlet energy minimization). *Fix a query node/state $s_t$ with current policy $\pi_t \in \Lambda_{s_t}$ and neighbors $\{\pi_j\}_{j \in \mathcal{N}_k(s_t)}$ with weights $\{w_{tj}\}_{j \in \mathcal{N}_k(s_t)}$. Define the* local *Dirichlet (memory) energy on $\Lambda_{s_t}$ by*

$$\Phi_t(\mu) := \frac{\lambda}{2} \sum_{j \in \mathcal{N}_k(s_t)} w_{tj} W_{2,\varepsilon}^2(\mu, \pi_j). \tag{114}$$

*Consider the (local) Wasserstein gradient flow*

$$\partial_\tau \mu_\tau = -\nabla^{W_{2,\varepsilon}} \Phi_t(\mu_\tau), \qquad \mu_\tau \in \Lambda_{s_t}. \tag{115}$$

*Then the implicit Euler discretization of equation 115 with step size $\tau > 0$ is given by the entropic Wasserstein proximal map*

$$\mu^+ \in \arg\min_{\mu \in \Lambda_{s_t}} \left\{ \frac{1}{2\tau} W_{2,\varepsilon}^2(\mu, \pi_t) + \Phi_t(\mu) \right\}. \tag{116}$$

*In particular, the OT-fuse update (with $\tau = 1$ in our implementation) can be interpreted as computing $\pi_t^{\mathrm{fuse}} \approx \mu^+$, i.e., one implicit Euler step that decreases the local Dirichlet energy. Moreover, by optimality of $\mu^+$,*

$$\Phi_t(\mu^+) + \frac{1}{2\tau} W_{2,\varepsilon}^2(\mu^+, \pi_t) \leq \Phi_t(\pi_t), \tag{117}$$

*so the fusion step is stabilizing: it cannot increase the local geometric energy and it stays within a controlled distributional displacement from $\pi_t$.*

# G   ADDITIONAL EXPERIMENTAL DETAILS

## G.1   PLT IN RL: ENVIRONMENTS INFORMATION AND EVALUATION PROTOCOLS

### G.1.1   PPO AND ATARI GAMES

In Xuance, the Atari benchmark consists of a collection of Atari 2600 environments simulated by Stella and interfaced through the Arcade Learning Environment (ALE). In this paper, we conduct comparative experiments on 20 representative tasks, with several examples illustrated in the Figure 5. The evaluation reward definitions are environment-specific, and the detailed specifications can be found in AtariAge.

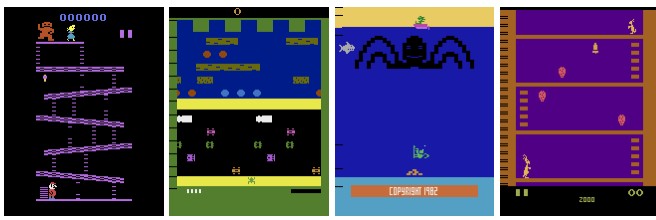

Figure 5: Atari Game Example: Donkey King, Frogger, Name This Game, Kangaroo

### G.1.2   MAPPO AND STARCRAFT II

The StarCraft Multi-Agent Challenge (SMAC) serves as WhiRL's benchmark environment for studying cooperative multi-agent reinforcement learning (MARL) algorithms. SMAC is built on StarCraft II, a real-time strategy game developed by Blizzard Entertainment, which provides a challenging and realistic testbed for cooperative control. In this paper, we conduct experiments on four combat maps, where performance is evaluated by the win rate, defined as the ability of the allied units to overcome the enemy forces.

## G.2   PLT IN LLM: DESIGN RATIONALE AND EVALUATION PROTOCOLS

**High-level goal.** We treat an LLM as a conditional policy that maps an input $x$ (prompt/state/context) to an action distribution over next tokens. Our LLM experiments are designed to stress three brittle failure modes of *LLMs-as-policies* that directly reflect the paper's claims: (i) **stability** under long-range, contradictory evidence, (ii) **robustness** under adversarial fact–rumor and credibility shifts, and (iii) **sample efficiency** when acquiring novel concepts without erasing existing ones. Across all experiments, we match training budgets and evaluation protocols between baselines and +PLT and report multi-seed statistics (mean $\pm$ std / 95% CI), so observed gains are unlikely to be artifacts of scaling or noise.

Our LLM evaluations are designed as controlled *stress tests* that isolate failure modes implied by the geometry-mismatch hypothesis, rather than to claim broad benchmark SOTA. **Counterfactual Trust** targets fact-verification and misinformation robustness under explicit fact–rumor conflicts and source-credibility shifts, aligning in spirit with established truthfulness/fact-checking evaluations such as *FEVER: A Large-scale Dataset for Fact Extraction and VERification* (Thorne et al., 2018) and *TruthfulQA: Measuring How Models Mimic Human Falsehoods* (Lin et al., 2022). **Long-Range Factual Recall** isolates distance-driven degradation in long-context use under controlled overwrites and distractors, complementing long-context benchmarks and analyses such as *Long-Bench: A Bilingual, Multitask Benchmark for Long Context Understanding* (Bai et al., 2024) and *Lost in the Middle: How Language Models Use Long Contexts* (Liu et al., 2024). Finally, **Few-Shot Learning** mirrors few-shot adaptation and continual-learning concerns by measuring novel-class acquisition under extreme scarcity while tracking base-class retention, connecting to in-context few-shot evaluation (Brown et al., 2020) and catastrophic-forgetting/continual-learning formulations (French, 1999). The advantage of these controlled tasks is that they hold confounds fixed (distance, conflict structure, and data scarcity), making them sensitive to update-trajectory stability rather than model scale alone.

### G.2.1 COUNTERFACTUAL TRUST TASK

**Motivation.** This task targets **decision stability** and **robustness** under a common real-world pathology: the model must make a *retrieval-conditioned decision* when evidence is (i) long-range, (ii) partially counterfactual, and (iii) adversarially conflicting (fact–rumor), with an additional *source credibility shift* dimension. The design intentionally separates "can the model retrieve/use evidence" from "does the model keep stable preferences under conflict."

**Task setup.** Each example consists of (a) a query requiring a trust/choice decision, (b) an in-context narrative containing both official and non-official claims, and (c) a retrieved context block placed at a controlled long distance (e.g., a 500-token retrieval window). We construct conditions that vary whether the decisive evidence is (i) present only in retrieval, (ii) contradicted by a rumor, and (iii) associated with a high-credibility or low-credibility source.

**Metrics and what they measure.** We report four complementary metrics:

- **General Acc.** Overall decision accuracy across all evaluation conditions. *Measures end-to-end policy correctness.*
- **Retrieval Acc.** Accuracy on the subset where the correct decision *requires* using retrieved evidence (e.g., when decisive facts appear only in the retrieval block at long range). *Measures long-range evidence integration as an action policy.*
- **Conflict Acc.** Accuracy under explicit fact–rumor contradictions (adversarial conflicts). *Measures robustness/stability under contradictory evidence.*
- **Official Acc.** Accuracy (or preference rate) for selecting the official/high-credibility source when credibility is the determining factor. *Measures whether the policy maintains stable credibility priors under distribution shift.*

**Scoring rule.** Each example is mapped to a discrete decision (e.g., select A/B, trust official vs rumor, etc.). A prediction is correct iff it matches the ground-truth decision under that condition. For **Retrieval Acc.** and **Conflict Acc.**, we evaluate only the pre-specified subsets (retrieval-required vs conflict-present), so the metric isolates the intended failure mode rather than being diluted by easy cases.

**Why this design supports our claims.** If PLT improves **geometry-aware stability** rather than merely fitting spurious patterns, we should observe: (i) larger gains on **Retrieval Acc.** than on easy in-context cases, and (ii) improved **Conflict Acc.** and **Official Acc.** when evidence is contradictory or credibility-shifted. These are precisely conditions where "small" parameter-space updates often produce disproportionate behavioral flips, and where using historical policies as a regularizer should stabilize policy preferences.

### G.2.2 LONG-RANGE FACTUAL RECALL (OVERWRITE & DISTRACTORS)

**Motivation.** This benchmark isolates **distance-driven degradation** (a stability failure) in a controlled setting: facts are repeatedly overwritten, and the model must retrieve the *latest* update at a known token distance under distractors. Unlike natural corpora, the overwrite structure gives an unambiguous ground-truth "final state" and makes failure modes attributable to long-range credit assignment rather than ambiguity.

**Task setup.** Each story contains at least two overwriteable variables (e.g., a moving object's location and a vault passcode). The final supporting update is inserted at controlled positions, inducing exact query–evidence token gaps. We optionally include distractors (irrelevant facts, misleading mentions, or outdated values) to stress robustness of late-evidence selection.

**Metrics and what they measure.** We score *answer spans only* (targets-only) to avoid conflating narrative modeling with factual retrieval:

- *Loss (Late)*: negative log-likelihood restricted to the final answer span when the decisive evidence is far from the query. *Measures whether late evidence is actually used.*

- *Move Acc / Pass. Acc*: exact-match accuracy for each overwritten variable. *Measures factual correctness under overwrite.*

- *Token Acc (W)*: length-weighted token-level accuracy on the answer span. *Measures partial correctness and robustness to answer length.*

- *NLL (Std)*: calibration/stability statistic across distance buckets (e.g., variance/dispersion of NLL). *Measures whether improvements come with unstable confidence or degraded calibration.*

**Scoring rule and distance buckets.** We group examples by query–evidence token distance (pre-defined buckets) and report both per-bucket and aggregate results. Targets-only scoring ensures improvements reflect *evidence retrieval and selection*, not generic language modeling.

**Why this design supports our claims.** A geometry-aware, experience-regularized update should reduce sharp behavioral drift that appears as "forgetting the latest overwrite" at long distance. Thus, the expected signature is lower *Loss (Late)* and higher accuracy at large token gaps, while maintaining (or improving) calibration (*NLL (Std)* not exploding). This directly operationalizes **stability** for LLM policies.

### G.2.3 FEW-SHOT LEARNING (BASE RETENTION + NOVEL ACQUISITION)

**Motivation.** Few-shot learning is a canonical **sample efficiency** test and also exposes **catastrophic forgetting**: we want fast acquisition of novel classes from extremely few examples *without* degrading performance on base classes.

**Task setup.** We use a controlled class-incremental protocol with 20 base classes (many labeled examples each) and 5 novel classes (few labeled examples each; e.g., 5-shot). Evaluation uses a mixed test set containing both base and novel classes, so improvements cannot come from simply shifting probability mass toward novel classes at the expense of base performance.

**Metrics and what they measure.**

- **Base Acc.**: accuracy on base classes after training with novel examples. *Measures retention / absence of catastrophic forgetting.*

- **Novel Acc.**: few-shot accuracy on novel classes. *Measures sample-efficient acquisition.*

- **Overall Acc.**: accuracy on the mixed test set. *Measures net utility under realistic mixture.*

**Scoring rule.** Predictions are correct iff the model outputs the correct class label (or canonicalized label string) under a fixed decoding scheme. We report multi-seed statistics and keep training budgets identical between baseline and +PLT.

**Why this design supports our claims.** If PLT is functioning as an experience-induced geometric regularizer on the policy trajectory, it should improve novel-class learning (higher **Novel Acc.**) *without* sacrificing the learned base geometry (stable **Base Acc.**). This is exactly the "sample efficiency without forgetting" signature.

## H EXPERIMENTAL SETUP AND COMPUTE.

All experiments are run on a single NVIDIA RTX 3090 (24GB). A typical run takes approximately 2 hours end-to-end under our default settings, while the peak GPU memory footprint varies with the backbone and policy parameterization (e.g., network width/depth, action distribution head, and batch size). Unless otherwise specified, we choose model sizes to fit within the 24GB budget and adjust architecture and batch/sequence lengths accordingly; for larger backbones or higher-capacity variants commonly used in top-tier benchmarks, we reduce batch size and/or sequence length (and, when needed, use gradient accumulation) to ensure comparable optimization dynamics under the same hardware constraints.

