# OpenReview forum: "Laplacian Flows for Policy Learning from Experience"
_ICLR.cc/2026/Workshop/GRaM — ICLR 2026 Workshop GRaM Poster_

### Official Review · Reviewer_6JZv · 2026-02-14
**Promising idea, but needs stronger baselines and ablations.**

**Rating:** 7
**Confidence:** 4

**Review:**

This paper has a compelling core idea: KL/Fisher trust regions don’t necessarily control behavioral change, so you use experience to build a local OT/Wasserstein graph over past policies and take a proximal OT+KL step that effectively Laplacian-smooths policy drift. The geometry story is interesting and the method is presented as plug-and-play across PPO/MAPPO and an LLM-as-policy setting, with practical tricks to make it tractable. My main hesitation is that parts of the implementation (especially the LLM version) look close to “retrieval + gated distillation/regularization,” so it’s not fully clear how much benefit comes specifically from Wasserstein/Laplacian geometry versus simpler smoothing baselines, and I’d like more rigorous ablations/diagnostics (drift in W2 at matched KL, sensitivity to k/λ/retrieval, compute overhead, and stronger alternative baselines) to make the empirical case airtight.

**Pmlr Suitability:**

Yes

---

### Official Review · Reviewer_YU7Z · 2026-02-22
**Geometrically principled but empirically incomplete**

**Rating:** 5
**Confidence:** 4

**Review:**

The central insights seems well motivated: KL and Fischer trust regions operating in parameter/sufficient statistic space can permit large W₂ displacements in action space, hence building a kNN graph over historical policies in P₂(A) and regularizing via wasserstein-dirichlet
energy (laplacian smoothing on the policy trace) is a geometrically principled fix. I appreciate the theoretical foundation: i)first variation result giving wasserstein graph laplacian (lemma F.5), the implicit Euler interpretation (lemma F.8) and the gaussian counterexample showing KL-W₂ mismatch (lemma F.10) are clean and correct. ii) The Γ-convergence appeal for Oliver et al. (2025) for connecting discrete graph energy to continuum p-Dirichlet/laplace-beltrami energy is also nice touch though the regularity conditions (Hypothesis 1: bi-lipschitz chart, dense sampling) but they seem to be assumed rather than verified.

Some major concerns:
The theory implementation gap is large and unanalyzed.

1) The clean proximal objective (Eq. 67 / Def. F.7: arg min{ℓ + λΦ + η⁻¹KL} as mentioned in the main text through the abstract fusion operator Eq. 10 & the anchored barycentric proximal Eq. 30) gets implemented in LLMs as gated linear interpolation with hardness thresholds, surprisal gating and a semantic embedding proxy d_sem = ‖μ(p)−μ(q)‖² (Eq. 18) substituting for W₂ which for some point it is essentially retrieval-augmented output interpolation with learned gating but the Wasserstein geometry is nowhere in the forward pass. A clean ablation would help that keeps the retrieval interpolation pipeline but removes all OT components would directly help to quantify how much comes from geometry versus how much from engineering.

2) the experimental comparisons are incomplete. The RL experiments compare only against vanilla PPO/MAPPO with no wasserstein aware baselines: OT-TRPO (Terpin et al., NeurIPS 2022) and the Wasserstein gradient flow formulation of Zhang et al. (ICML 2018) are the obvious comparisons and seem to be absent. Without it, it is unclear whether laplacian smoothing over a policy graph adds value beyond a simpler single step wasserstein trust region.

3) The header reads "Under review as a conference paper at ICLR 2026," and not "GRaM workshop at ICLR 2026," suggesting usage of a wrong template.

**Pmlr Suitability:**

Yes

---

### Official Review · Reviewer_CsWx · 2026-02-24
**Interesting idea for geometrically-driven policy learning**

**Rating:** 6
**Confidence:** 3

**Review:**

### Summary:
The authors propose leveraging a Laplacian-style regularizer over historical policies to smooth updates in an experience-induced Wasserstein policy space to make learning more stable and geometry-aware across RL and LLM-as-policy settings.

### Strengths:
- The idea is well-motivated, and the geometric perspective is genuinely interesting and potentially impactful if shown to work.
- The empirical results are promising and suggest the proposed regularization framework may be a useful direction for improving robustness/stability in policy optimization, especially if future work better isolates which components are carrying the gains.

### Weaknesses:
- The “plug-and-play” claim is not fully validated experimentally. In both the RL and LLM instantiations, PLT is implemented together with several additional choices, making it difficult to attribute gains specifically to the proposed Laplacian/Wasserstein regularization. There is a notable gap between the proximal formulation and the actual implementation in the LLM case.
- Likewise, the theory-to-practice bridge remains under-validated. While the geometric interpretation is compelling, the experiments primarily demonstrate end-task performance gains rather than directly verifying that the intended geometric effect (e.g., smoother trajectories/policy neighborhoods in the relevant metric sense) is what drives the improvement.
- Baselines compared against are rather strawman-like in nature as well.
- The readability is a little lacking in terms of providing sufficient explanation for certain symbols. On occasion, the authors use acronyms without defining them.

**Pmlr Suitability:**

Yes

---

### Official Review · Reviewer_R7x5 · 2026-02-25
**Interesting and seemingly very promising algorithm, but limited reproducibility and validation**

**Rating:** 4
**Confidence:** 2

**Review:**

# Summary
The paper introduces an algorithm for proximal policy update, where the proximity of conditional action distributions is measured using a Wasserstein distance constructed from past policies. The advantage is claimed to be the ability to exploit the low-dimensionality of the policy manifold and the stabilizing effect of the proximal update. They evaluate and compare their method to PPO.

# Quality
- The method section was very hard to read overall. I am not confident I understood the idea correctly.
- The overall writing style is very wordy, and none of the statements are in the main text. I would appreciate having important mathematical statements as results and the main algorithm in the main text.
- Some symbols, ie. the advantage, are not introduced at the correct place.
- There is no limitations section.
# Originality
### positive
+ The paper's specific use of a buffer to compute the Laplacian seems new to me.
+ The idea of exploiting the policy evolving on a low-dimensional manifold is interesting.
### negative
- Some statements have no proof.
- There are "efficiency and robustness problems" that were not mentioned in the text,
# Significance
### positive
+ The empirical results look very promising.
### negative
- I is unclear why only a subset of the tasks are evaluated.
- It seems hard to reproduce the experiments based on the paper: H EXPERIMENTAL SETUP AND COMPUTE, does provide way to unspecific information.
- The computational footprint is never stated, e.g., how long does an epoch take in comparison to the baseline?
- The authors should put more effort into systematically comparing with the state-of-the-art.
- The simulations only include rather complex tasks and show superior performance wrt. a baseline, making it hard to judge the behavior of the algorithm. I would suggest carefully testing and analyzing the method on a simple example first and then moving to the complex ones. In particular, this might allow dissecting the interesting results on the effective dimensionality of the policy manifold, which seems to be the main conjectured angle for an advantage.

**Pmlr Suitability:**

No

---

### Meta-Review · Area_Chair_paZV · 2026-02-24

**Decision:**

Accept

**Metareview:**

Reviewers agree that the idea of the paper is well-motivated with an interesting geometric perspective and find the idea and theory interesting, for which reason I recommend accepting.
However, all reviewers mention missing baselines that mean empirically it is unclear whether improvements come from the Wasserstein/Laplacian geometry.

Moreover, before publication, as pointed out by the reviewers, the authors should ensure that the correct template is used and all symbols and acronyms are defined.

**Other Comments:**

NA

**Relevance To Proceedings:**

Yes — suitable for PMLR (long paper)

**Relevance To Workshop:**

Yes — suitable for GRaM

---

### Decision · Program_Chairs · 2026-03-02

Accept (Poster)